# SONAR: Long-Range Graph Propagation Through Information Waves

**Alessandro Trenta**[*]
Department of Computer Science
University of Pisa
Pisa, Italy
alessandro.trenta@phd.unipi.it

**Alessio Gravina**[*]
Department of Computer Science
University of Pisa
Pisa, Italy
alessio.gravina@di.unipi.it

**Davide Bacciu**
Department of Computer Science
University of Pisa
Pisa, Italy
davide.bacciu@unipi.it

## Abstract

Capturing effective long-range information propagation remains a fundamental yet challenging problem in graph representation learning. Motivated by this, we introduce SONAR, a novel GNN architecture inspired by the dynamics of wave propagation in continuous media. SONAR models information flow on graphs as oscillations governed by the wave equation, allowing it to maintain effective propagation dynamics over long distances. By integrating adaptive edge resistances and state-dependent external forces, our method balances conservative and non-conservative behaviors, improving the ability to learn more complex dynamics. We provide a rigorous theoretical analysis of SONAR's energy conservation and information propagation properties, demonstrating its capacity to address the long-range propagation problem. Extensive experiments on synthetic and real-world benchmarks confirm that SONAR achieves state-of-the-art performance, particularly on tasks requiring long-range information exchange.

## 1 Introduction

Graph neural networks (GNNs) [67, 76, 10, 18] have become a powerful framework for processing graph-structured data, enabling applications across various domains such as social networks [59, 49], molecular biology [36, 89, 43, 5], and more [38, 58]. Most GNNs are built upon the Message-Passing Neural Networks (MPNNs) framework [36], where information is exchanged between neighboring nodes, enabling effective learning from local graph structure. Despite their widespread use and success, a persistent challenge in GNNs is the effective modeling of long-range dependencies within graphs, as information propagation tends to degrade over extended distances. This is caused by phenomena such as over-smoothing [11, 73], over-squashing [2, 19], and, more generally, vanishing gradients [3]. Several recent approaches have been developed to better capture long-range dependencies in graphs, including graph rewiring [35, 83, 7], multi-hop GNNs [1, 46, 20], differential-equation-based GNNs (DE-GNNs) [13, 39, 40, 52], and graph transformers [78, 23, 90, 72, 88]. Graph transformers, in particular, have gained popularity due to their use of attention mechanisms, which allow any pair of nodes to exchange information directly. However,

---

[*]Equal contribution

their quadratic computational cost poses scalability challenges. Moreover, recent studies have shown that, on some long-range tasks, they may actually perform worse than standard MPNNs [82].

To address the long-range propagation problem, we draw inspiration from wave propagation in continuous media, where signals can travel vast distances with minimal energy loss [29]. We propose **SONAR** (**S**tructured **O**scillatory Graph Neural **N**etwork with **A**daptive **R**esistance), a novel DE-GNN with a MPNN-like architecture that models information propagation through the lens of wave-based dynamics. Unlike diffusion-based propagation schemes such as [59, 85, 89], which are inherently dissipative and struggle to capture long-range interactions [3], wave-based dynamics preserve signal energy and enable effective communication across distant nodes. This mechanism is analogous to acoustic wave propagation in the ocean, where, under specific temperature and pressure gradients, sound waves can become trapped in the SOFAR channel [30]. This natural waveguide allows acoustic signals to travel across thousands of kilometers with minimal attenuation. SONAR adopts a similar principle in the graph domain: by treating information as a wave propagating through the graph topology, it maintains signal fidelity over many hops and facilitates robust long-range interactions. Specifically, it adopts a mathematically grounded formulation to model wave dynamics [34], resulting in inherently linear propagation dynamics with strong theoretical guarantees. Moreover, SONAR introduces adaptive resistances that enable each edge to control the flow of information, along with external forces that balance conservative and non-conservative dynamics, allowing the model to capture more complex behaviors and improve performance on downstream tasks.

The key contributions of this work are the following: (i) We propose SONAR, a novel MPNN-like architecture based on the wave equation for graphs, which enables the balance and integration of non-dissipative long-range propagation and non-conservative behaviors. Additionally, it incorporates adaptive resistances that allow each edge to modulate the information flow. (ii) We theoretically prove that, in its conservative form, SONAR does not dissipate the energy of features in the graph, i.e., information is preserved. In its continuous form, the sensitivity matrix between any two nodes never vanishes but oscillates following a cosine function. (iii) We employ additional dissipative and forcing terms, allowing SONAR to mediate between pure conservative and non-conservative behaviors. This gives SONAR the flexibility to learn to filter out irrelevant information. (iv) We conduct extensive experiments to demonstrate the benefits of our method. SONAR consistently achieves on par or better performance than existing methods across diverse tasks and benchmarks. Notably, SONAR outperforms existing state-of-the-art methods on synthetic long-range benchmarks, where accurate modeling of distant node interactions is crucial, highlighting its strength in long-range propagation.

## 2 Related Work

**Long-Range Propagation in GNNs.** Effectively modeling long-range dependencies remains a central challenge in deep learning for graphs [77, 3]. GNNs usually rely on local neighborhood aggregation, which limits their capacity to capture interactions between distant nodes [2, 19] due to challenges such as over-smoothing [11, 68, 73] and over-squashing [2, 83, 19], which are linked to the problem of vanishing gradients [3]. Therefore, GNNs based on message passing exhibit a performance degradation in tasks requiring more global context, such as molecular property prediction [24]. To address these challenges, a variety of strategies have been proposed. Graph rewiring methods, including SDRF [83], DIGL [35], FoSR [57], and DRew [46], modify the graph topology to facilitate long-range communication. Other approaches include regularizing the model's weight space [39, 41, 40], exploiting port-Hamiltonian dynamics [52], filtering messages in the information flow [28, 32], using a graph adaptive method based on a learnable ARMA framework [25], or using multi-hop information in a single update [21]. Graph Transformers [61, 72, 78, 80, 90] offer an alternative paradigm by enabling direct message passing between any pair of nodes via attention mechanisms. While effective, most of these methods often introduce additional computational complexity due to denser graph shift operators or all-pairs interactions.

**GNNs based on Differential Equations.** Recent advancements in the field of representation learning have introduced new architectures that establish a connection between neural networks and dynamical systems. Building on foundational work in recurrent neural networks [16, 47], this perspective has been extended to GNNs [50]. Indeed, works like GDE [71], GRAND [13], PDE-GCN [26], DGC [87], GRAND++ [81] propose to interpret GNNs as discretisations of ODEs and PDEs. Methods in this class impose a bias on node representation trajectories to follow the heat diffusion process [13, 81, 87], exploit non-dissipative dynamics [39, 40], or hamiltonian dynamics [56, 91, 52].GraphCON [74]

further explores oscillatory dynamics to preserve the Dirichlet energy encoded in the node features and mitigate oversmoothing. In GraphCON, each node acts as an oscillator that exchanges information with its neighbors through a simple GCN [59] or GAT [85]. In contrast, SONAR adopts a mathematically grounded formulation of graph calculus [33] to model wave propagation on graphs [34], resulting in inherently linear dynamics that offer stronger theoretical guarantees for long-range information propagation. Moreover, SONAR introduces adaptive resistances (which allow each edge to modulate how information is transmitted) as well as external forces that enable the modeling of more complex dynamics. The spatio-temporal evolution of graphs has been studied in [27, 42, 55, 44].

## 3 SONAR

In this section, we introduce our **SONAR** (**S**tructured **O**scillatory Graph **N**eural **N**etwork with **A**daptive **R**esistance), a novel GNN architecture whose information flow is designed as the wave equation on graphs, enabling long-range information propagation between nodes. Figure 1 provides a visual representation of the overall architecture and wave propagation dynamics.

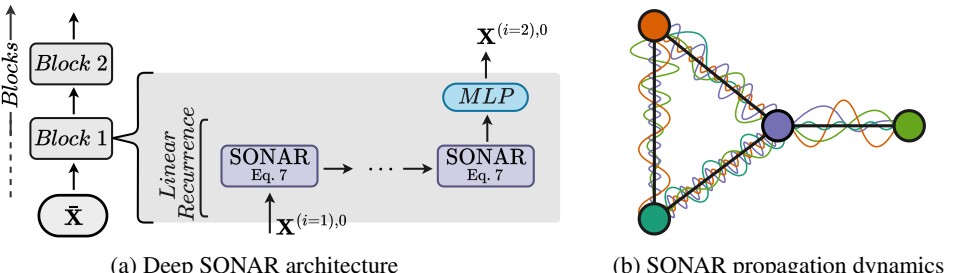

Figure 1: Illustration of (a) a deep SONAR architecture and (b) the SONAR propagation dynamics of one block.

### 3.1 Notations and Preliminaries

We consider a graph $\mathcal{G} = (\mathcal{V}, \mathcal{E})$ as the system of interacting entities called nodes, where $\mathcal{V}$ is a set of $n$ nodes and $\mathcal{E} \subseteq \mathcal{V} \times \mathcal{V}$ is a set of $m$ edges. The structural information expressed by $\mathcal{E}$ can be encoded in the adjacency matrix $\mathbf{A} \in \{0,1\}^{n \times n}$, i.e., a binary matrix where $\mathbf{A}_{ij} = 1$ if $(i,j) \in \mathcal{E}$ and zero otherwise. We define the neighborhood of node $v$ as the set of nodes $u$ directly linked with $v$, i.e., $\mathcal{N}(v) = \{u | \mathbf{A}_{uv} = 1\}$, and the degree of node $v$ as the number of its neighbors, $\deg(v) = |\mathcal{N}(v)|$. Lastly, we consider the Laplacian matrix $\mathbf{L}$ and two normalizations, i.e., symmetric and random-walk normalized Laplacian matrices, which are respectively defined as

$$\mathbf{L} = \mathbf{D} - \mathbf{A}, \qquad \mathbf{L}^{\text{sym}} = \mathbf{I} - (\mathbf{D}^+)^{\frac{1}{2}} \mathbf{A} (\mathbf{D}^+)^{\frac{1}{2}}, \qquad \mathbf{L}^{\text{rw}} = \mathbf{I} - \mathbf{D}^+ \mathbf{A} \tag{1}$$

where $\mathbf{I}$ is the identity matrix, $\mathbf{D}$ is the diagonal degree matrix with $\mathbf{D}_{ii} = \deg(v_i)$ and $v_i$ a generic node. $\mathbf{D}^+$ is the pseudo-inverse of $\mathbf{D}$. Following [34], the definition of graph Laplacian can be generalized to its weighted version and to any node function $f(v) : \mathcal{V} \to \mathbb{R}$ as

$$(\Delta f)(v) = (\mathbf{L}^a f)(v) = (\mathbf{D}^a f)(v) - (\mathbf{A}^a f)(v) = \sum_{u \in \mathcal{N}(v)} a_{uv}(f(v) - f(u)) \tag{2}$$

where $\mathbf{D}^a$ and $\mathbf{A}^a$ are the weighted degree and adjacency matrices, and $a_{uv} \in \mathbb{R}_+$ is a weight on the directed edge $(u, v) \in \mathcal{E}$.

Each node in the graph is associated with a hidden state vector $\mathbf{x}_v(t) \in \mathbb{R}^d$, which provides the evolution of the node representation at time $t$ in the system. The term $\mathbf{X}(t) = [\mathbf{x}_0(t), \dots, \mathbf{x}_{n-1}(t)]^\top \in \mathbb{R}^{n \times d}$ is the matrix of all node states at time $t$. We note that in the domain of differential-equations the continuous dynamics expressed by the neural network is discretized into classical GNN layers [39]. Therefore, given the initial condition (node input features) $\mathbf{x}_v(0) = \bar{\mathbf{x}}_v$, each GNN layer $\ell = 1, \dots, L$ computes node states $\mathbf{x}_v^\ell$ which approximate $\mathbf{x}_v(t = h\ell)$, with $h$ being the integration step size.

**The wave equation in physics.** Waves in physics describe the propagation of oscillations in a continuous medium. Common examples are electromagnetic waves [62] and sound waves [48]. These are described by the wave equation $\frac{\partial^2 f}{\partial t^2} = c^2 \frac{\partial^2 f}{\partial x^2}$, where $f(t, x)$ is a scalar function describing the wave and $c$ is its propagation speed, which is an intrinsic characteristic of the conditions and medium. Directly descending from their equation, waves have interesting properties: (i) they can travel indefinitely without losing amplitude and energy at constant speed [29], and (ii) they can be linearly composed and decomposed, allowing for superposition of waves propagating from different directions [62]. More specifically, the total energy $E(t)$ of a wave on $\mathbb{R}$ is defined as

$$E(t) = \int_{\mathbb{R}} \left( \frac{\partial f}{\partial t} \right)^2 + \left( \frac{\partial f}{\partial x} \right)^2 dx, \tag{3}$$

and can be shown to be constant [29] in the whole space and even on bounded domains. Our objective is to exploit these properties to propagate energy and information indefinitely a graph.

Waves following a non-conservative dynamic can be modeled with the addition of a dissipative term $D(t, x) = k\frac{\partial f}{\partial t}(t, x)$, which dampens the wave and dissipates energy, or with an external forcing term $F(t, x)$, which represents other environmental factors acting on the wave.

## 3.2 The SONAR wave propagation

To model information propagation on graphs as oscillations in a continuous medium, we formulate the dynamics of SONAR through a differential equation that follows the graph wave equation. Specifically, we start by defining the evolution of node states as:

$$\ddot{\mathbf{X}}(t) = -\mathbf{L}\mathbf{X}(t)\mathbf{W}, \qquad \mathbf{X}(0) = \bar{\mathbf{X}}, \tag{4}$$

where $\ddot{\mathbf{X}}(t)$ is the second-order derivative of $\mathbf{X}(t)$, $\mathbf{W} \in \mathbb{R}^{d \times d}$ is a learnable weight matrix, and $\bar{\mathbf{X}}$ contains the initial node states. To more closely mimic wave propagation in physical media, we adopt the weighted graph Laplacian in Equation (2), allowing different edges to modulate how signals travel through the graph. In other words, each pairwise communication between nodes can be viewed as residing in a distinct medium, where the wave properties, e.g., speed and amplitude, vary depending on the properties of the medium. This is analogous to propagating signals at different depths in the ocean, where specific water temperature and pressure alter the wave propagation capabilities. Therefore, in our setting, the Laplacian weight $a_{uv}$ can be interpreted as an inverse resistance term that governs the ease with which information flows between nodes $u$ and $v$. Although $a_{uv}$ can be implemented as static (fixed) weights, in our experiments in Section 4 we employ neural networks to learn such weights in order to adaptively model the dynamics of the propagation on the specific task.[2]

To further cast the system dynamics in the more general setting, we then introduce the possibility of trading between the conservative behaviour typical of undampened wave equations (discussed in detail in Section 3.3) with non-conservative dynamics. We introduce dissipative and external forcing terms to alter the conservative evolution of node states. Therefore, Equation (4) can be rewritten to include the new terms as

$$\ddot{\mathbf{X}}(t) = -\mathbf{L}^a \mathbf{X}(t)\mathbf{W} - D(\mathbf{X}(t)) \odot \dot{\mathbf{X}}(t) + F(\mathbf{X}(t)), \tag{5}$$

where $D(\mathbf{X}(t)) \in \mathbb{R}_+^{n \times d}$ and $F(\mathbf{X}(t)) \in \mathbb{R}^{n \times d}$ denote the state-dependent dissipative and external forcing terms, respectively. $\odot$ is the Kronecker product. We note that without $D(\mathbf{X}(t))$ and $F(\mathbf{X}(t))$, the purely conservative inductive bias of the wave equations constrains node states to evolve along conservative trajectories, which may limit the model's effectiveness and hinder its ability to capture the more complex dynamics required for downstream tasks. Therefore, the dissipative term allows irrelevant information to be dissipated during propagation, while the external force introduces additional flexibility to control the dynamics based on input-dependent signals. In our experiments in Section 4, we employ neural networks to learn both the dissipative and external forcing terms.

**Discretization and implementation details.** As for standard DE-GNNs, a numerical discretization method is needed to solve Equation (5). We solve the equation with a finite difference scheme [31].

---

[2]Note that disconnected nodes always have a zero adaptive resistance. Thus, the resistance $a_{uv}$ is calculated only between connected nodes.

To this aim, we first introduce the auxiliary velocity variable to be the first-order derivative of the node states, i.e., $\mathbf{V}(t) = \dot{\mathbf{X}}(t)$, to rewrite Equation (5) as the first-order system:

$$\begin{cases} \dot{\mathbf{X}}(t) = \mathbf{V}(t) \\ \dot{\mathbf{V}}(t) = -\mathbf{L}^a \mathbf{X}(t) \mathbf{W} - D(\mathbf{X}(t)) \odot \dot{\mathbf{V}}(t) + F(\mathbf{X}(t)). \end{cases} \tag{6}$$

This system can be now discretized by iteratively solving

$$\begin{cases} \mathbf{X}^{\ell+1} = \mathbf{X}^\ell + h\mathbf{V}^{\ell+1}, \\ \mathbf{V}^{\ell+1} = \mathbf{V}^\ell - h\left(\mathbf{L}^a \mathbf{X}^\ell \mathbf{W} + D(\mathbf{X}^\ell) \odot \mathbf{V}^\ell - F(\mathbf{X}^\ell)\right), \end{cases} \tag{7}$$

where $h$ is the step size of the discretization method, and $\mathbf{X}^0 = \bar{\mathbf{X}}$ and $\mathbf{V}^0 = \mathbf{X}^0 \mathbf{W}_V$, with $\mathbf{W}_V \in \mathbb{R}^{d \times d}$ a learnable matrix for the initial velocity. Note that edge features can be seamlessly incorporated into Equation (7), as detailed in Appendix A.1. As discussed above, we can discretize time into GNN layers, such that the embedding $\mathbf{x}_v^\ell$ from Equation (7) encodes information that is $\ell$ hops away from $v$.

We note that, when dissipation and external forcing are not employed, information is linearly aggregated. To increase effectiveness, we can build a deep architecture for our SONAR, following principles established in modern architectures [69, 45, 12]. Therefore, we define a SONAR block by applying an MLP to the output of Equation (7) after $L$ propagation steps, introducing nonlinear dynamics into the model. Multiple SONAR blocks can then be stacked to form a deep architecture, where the output of block $i$ becomes the initial condition of the subsequent block $(i + 1)$:

$$\begin{aligned} \mathbf{X}^{(i),L} &= \text{SONAR}(\mathbf{X}^{(i),0}) \\ \mathbf{X}^{(i+1),0} &= \text{MLP}(\mathbf{X}^{(i),L}). \end{aligned} \tag{8}$$

This concept is visually summarized in Figure 1a.

In our experiments in Section 4, the Laplacian weights $a_{uv}$ are implemented as the output of an MLP that takes the states $\mathbf{x}_v^\ell$ and $\mathbf{x}_u^\ell$ as input, followed by a ReLU activation to ensure a positive output. The dissipative term is modeled using an MLP applied to the current node state, also followed by a ReLU to enforce non-negative outputs. The external forcing term is computed via an MLP without activation constraints, allowing it to output both positive and negative values. As a result, each propagation step (i.e., iteration of the discretization step) $\ell$ in each block is associated with its own set of learned Laplacian weights, as well as dissipative and external forces. Each SONAR block is equipped with its own set of parameters for learning the Laplacian weights, the dissipative and external forcing components, as well as its own learnable matrices for initial velocity $\mathbf{W}_V^{(i)}$, and message passing aggregation weights $\mathbf{W}^{(i)}$ (which are shared within the block). We consider both the number of blocks and propagation steps as hyperparameters in our experiments.

### 3.3 Theoretical Properties of SONAR

We now provide theoretical statements about energy and information conservation properties of SONAR, showing that our model effectively performs long-range propagation between nodes. Appendix B provides the proofs for the statements.

**SONAR allows for long-range propagation.** To prove that SONAR has a conservative behavior, we start by defining the energy of the system, adapting Equation (3) to the graph domain, as

$$E(t) = \underbrace{\sum_{v \in V} \frac{1}{2} \|\nabla \mathbf{x}_v(t)\|^2}_{\text{potential energy}} + \underbrace{\frac{1}{2} \left\| \frac{\partial \mathbf{X}(t)}{\partial t} \right\|^2}_{\text{kinetic energy}}. \tag{9}$$

The energy contains two components: the potential energy, which measures how much the signal varies across the graph (i.e., spatial variation)[3]; and the kinetic energy, which captures how node states change over time (i.e., layer-wise variation). In other words, the kinetic energy tells how fast the wave is vibrating, while the potential energy reflects the tension in the system. By maintaining a constant energy, we therefore ensure that node information is preserved during propagation.

---

[3]The potential energy is equivalent with the Dirichlet energy in [37] employed to measure over-smoothing.

To better highlight how node coupling and SONAR convolution affect energy conservation in the graph, we now focus on a single feature $\mathbf{X}(t) \in \mathbb{R}^n$. The generalization to multiple features is left to Appendix B. The following theorem shows that without dissipation and external forcing (i.e., with a propagation similar to Equation (4)), the wave propagation through the graph conserves the energy.

**Theorem 3.1.** *Let* $\mathbf{X}(t) \in \mathbb{R}^n$ *be the node states at time* $t$, *obtained as the solution to the graph wave equation in Equation* (5), *with initial condition* $\mathbf{X}(0) = \bar{\mathbf{X}}$, *null dissipative and external forcing terms. Then, the energy* $E(t)$ *in Equation* (9) *is conserved, that is,* $E(t) = E(0) \ \forall t > 0$.

This result implies that information encoded in the node states is neither amplified nor dissipated over time (i.e., while traversing the graph), but redistributed in a lossless manner across the graph. This property enables SONAR to perform deeper propagation, making it particularly suitable for tasks requiring long-range interactions.

To provide a clearer picture of the long-range capabilities of our SONAR, we follow the recent literature [83, 19] and evaluate the long-range propagation ability by measuring the sensitivity of the node states. Specifically, we first measure how sensitive is a node state at an arbitrary time $t$ with respect to the initial state of another node, i.e., $\partial \mathbf{x}_v(t) / \partial \mathbf{x}_u(0)$. Then, we compute the same sensitivity for the discretization version of our SONAR, i.e., $\partial \mathbf{x}_v^\ell / \partial \mathbf{x}_u^0$, and compute its norm similarly to [19]. We consider SONAR with null dissipative and external forcing terms, and initial conditions $\mathbf{X}(0) = \bar{\mathbf{X}} \in \mathbb{R}^n$, $\mathbf{V}(0) = \mathbf{0}$. The explicit solution of our system (following [34]) is

$$\mathbf{X}(t) = \cos\left(t\sqrt{\mathbf{L}}\right) \bar{\mathbf{X}}, \tag{10}$$

where $\cos\left(t\sqrt{\mathbf{L}}\right) = \sum_{n=0}^{\infty}(-1)^n \frac{t^{2n}}{n!} \sqrt{\mathbf{L}}^{2n}$. We drop the Laplacian superscript $^a$ to ease notation and explicitly write the sensitivity matrix in the following theorem.

**Theorem 3.2** (Sensitivity matrix, continuous case)**.** *Let* $\mathbf{x}_v(t)$ *be the state for node* $v$ *at time* $t$*. Then, the sensitivity* $\frac{\partial \mathbf{x}_v(t)}{\partial \mathbf{x}_u(0)}$ *between nodes* $u$ *and* $v$ *is*

$$\frac{\partial \mathbf{x}_v(t)}{\partial \mathbf{x}_u(0)} = \cos\left(t\sqrt{\mathbf{L}}\right)_{vu}. \tag{11}$$

The proof is a consequence of the solution in Equation (10). Theorem 3.2 shows that the influence of node $u$ on node $v$ oscillates following a cosine function, but never vanishes definitively.

We now consider a full discrete message passing with state vectors $\mathbf{X}^\ell \in \mathbb{R}^{n \times d}$ and measure the sensitivity after $\ell$ steps (i.e., layers).

**Theorem 3.3** (One-step sensitivity matrix, discrete case)**.** *The sensitivity matrix* $\frac{\partial \mathbf{x}_v^\ell}{\partial \mathbf{x}_u^{\ell-1}} \in \mathbb{R}^{d \times d}$ *of Equation* (7) *with null dissipative and external forcing terms is given by*

$$\frac{\partial \mathbf{x}_v^\ell}{\partial \mathbf{x}_u^{\ell-1}} = 2\mathbf{I}_{uv} - h^2 \mathbf{L}_{uv}\mathbf{W}. \tag{12}$$

The step size $h$ balances two components: the residual information from the node itself, related to the term $2\mathbf{I}_{uv}$, and the signal coming from neighboring nodes, related to $\mathbf{L}_{uv}\mathbf{W}$. This last term is also proportional to the difference between node features (see the definition of the graph Laplacian in Equation 2), encouraging the exchange of information when neighboring nodes are different.

Similarly to [19], we now assess sensitivity of our SONAR in Equation (7) for the full non-conservative case, i.e., with dissipation and external forcing.

**Theorem 3.4** (Sensitivity bound, discrete case)**.** *Consider the SONAR in Equation* (7) *on node states* $\mathbf{X}(t) \in \mathbb{R}^{n \times d}$*. Let the dissipation coefficient be such that* $|D(\mathbf{X}^\ell)| \leq k$ *and the external forcing be* $F(\mathbf{X}^\ell) = \mathbf{X}^\ell \mathbf{W}_F$*. Let the initial velocity be calculated as* $\mathbf{V}(0) = \bar{\mathbf{X}}\mathbf{W}_V$*. Finally, let* $w = \max\{|\mathbf{W}|, |\mathbf{W}_F|, |\mathbf{W}_V|, 1\}$*. Then, the sensitivity matrix has the following upper bound*

$$\left\| \frac{\partial \mathbf{x}_v^\ell}{\partial \mathbf{x}_u^0} \right\|_{L_1} \leq (wd)^\ell \left( \left( (1 + h + h^2(N + k + 1))\mathbf{I} + h^2\mathbf{A} \right)^\ell \right)_{vu} \tag{13}$$

*where* $N = \max_{v \in V} N_v = \max_{v \in V} |\mathcal{N}(v)|$ *is the maximum degree in the graph,* $d$ *is the number of node features, and* $h$ *is the step size of the discretization.*

This result demonstrates that the sensitivity of SONAR remains well-controlled across layers. We note that classical MPNNs usually include a factor $c_\sigma^\ell$ in the bound, with $c_\sigma$ the Lipschitz constant of the nonlinearity $\sigma$. In practice, this term often decay extremely fast, limiting the ability of standard MPNNs to propagate information over long distances. By contrast, the linear propagation dynamics of our SONAR allows for stable long-range information flow. Therefore, together with the previous theoretical results, it holds the capability of SONAR to perform long-range propagation effectively.

We remark that Theorem 3.1 and Theorem 3.2 do not depend on the choice of the discretization of the solution, while the results in Theorem 3.3 and Theorem 3.4, which explicitly depend on the step size, can be easily extended to any other integration procedure.

**Complexity Analysis.** SONAR consists of a stack of blocks, each with complexity of an MPNN (e.g., [59, 89]). Specifically, each iteration of Equation (7) is linear in the number of nodes ($n$) and edges ($m$), therefore it has a complexity of $\mathcal{O}(n + m)$. Assuming that $L$ iterations are performed, a SONAR block has a complexity of $\mathcal{O}(L(n + m) + \rho)$, where $\rho$ is the complexity of the MLP at the end of the block, as defined in Equation (8).

## 4 Experiments

In this section, we empirically validate the practical benefits of our method on popular graph benchmarks for long-range propagation as well as heterophilic node classification tasks. In Sections 4.1 and 4.2, we assess SONAR on synthetic benchmarks that require the exchange of messages between far-away nodes, thus performing long-range propagation. Specifically, we consider the graph transfer tasks from [40] and the task of predicting three graph properties from [39]. With the same purpose, we verify our method on the real-world long-range graph benchmark [24] in Section 4.3. Moreover, we assess the performance of SONAR on heterophilic tasks from [70] in Section 4.4. In Section 4.5, we empirically assess the long-range capabilites of SONAR in terms of the sensitivity metric (discussed in Section 3.3). In Appendix C.2, we report additional ablation studies to provide a more comprehensive understanding of SONAR, discussing runtimes and the role of adaptive resistance, dissipation, external forces, and step size. The performance of SONAR is compared with state-of-the-art methods, such as MPNNs, DE-GNNs, higher-order GNNs, and graph transformers, detailing the employed baselines in Appendix A.2. We report the hyperparameter space used in our experiments in Appendix A.4. All the experiments are performed on a server with NVIDIA H100 GPUs. We openly release the code at `https://github.com/gravins/SONAR`.

### 4.1 Graph Transfer Task

**Setup.** We consider the graph transfer task proposed by [19] under the experimental setting of [40]. The objective of this experiment is to transfer a label from a source to a target node placed at increasing distance $\ell$, and measure how much information is propagated through the graph. We initialize nodes with a random valued feature, and we assign values "1" and "0" to source and target nodes, respectively. We consider three graph distributions, i.e., line, ring, crossed-ring, and four different distances $\ell = \{3, 5, 10, 50\}$. Thus, we consider short to extreme long-range scenarios. As $\ell$ increases, the task becomes progressively more challenging, demanding more effective long-range information propagation. Due to oversquashing, the performance is expected to deteriorate with larger $\ell$. Consequently, addressing this problem requires methods with increasingly robust mechanisms for maintaining information flow across distant nodes. For this experiment, we use the same data, hyperparameter space, and experimental setting of [40].

**Results.** The results of SONAR and baseline models on the graph transfer task are shown in Figure 2, where we report the test mean squared error with standard error bars as a function of the distance between the source and target nodes. This task is specifically designed to evaluate a model's ability to propagate information across varying distances, making it a critical benchmark for assessing long-range capabilities. SONAR clearly outperforms all baselines at large distances (most notably at 50 hops) empirically validating its ability to support long-range information propagation. Moreover, SONAR achieves state-of-the-art performance across all tested distances, including shorter settings such as 3, 5, and 10 hops. Notably, while other models exhibit significant degradation as the distance increases, SONAR maintains a nearly constant error, demonstrating that it can propagate information effectively over both short and long ranges with consistent accuracy.

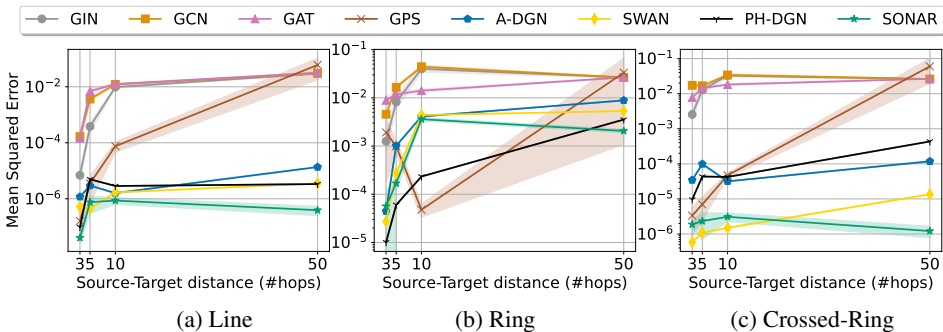

|  | (a) Line | (b) Ring | (c) Crossed-Ring |

Figure 2: Information transfer performance on (a) Line, (b) Ring, and (c) Crossed-Ring graphs. Overall, SONAR transfers the information more accurately as distance increases, achieving a lower error than non-dissipative methods (i.e., A-DGN, SWAN, PH-DGN) and transformers (i.e., GPS).

## 4.2 Graph Property Prediction

**Setup.** We consider the three graph property prediction tasks proposed in [39] and investigate the performance of our SONAR in predicting graph diameter, single source shortest path (SSSP), and node eccentricity on synthetic graphs sampled from multiple distributions. These tasks inherently demand the ability to capture and transmit information across the graph, as they are based on shortest-path computations. Indeed, similarly to standard algorithmic approaches (e.g., Bellman-Ford, Dijkstra's algorithm), accurate solutions depend on the exchange of multiple messages between nodes. Consequently, models that struggle with long-range propagation result in poor performance. For this experiment, we use the same data, hyperparameters space, and experimental setting presented in [39].

Table 1: Mean test set $log_{10}(\text{MSE})(\downarrow)$ and std averaged on 4 random weight initializations on Graph Property Prediction tasks. The lower, the better. First, second, and third best results for each task are color-coded.

| Model | Diameter | SSSP | Eccentricity |
|---|---|---|---|
| **MPNNs** | | | |
| GCN | $0.7424_{\pm0.0466}$ | $0.9499_{\pm0.0001}$ | $0.8468_{\pm0.0028}$ |
| GAT | $0.8221_{\pm0.0752}$ | $0.6951_{\pm0.1499}$ | $0.7909_{\pm0.0222}$ |
| GraphSAGE | $0.8645_{\pm0.0401}$ | $0.2863_{\pm0.1843}$ | $0.7863_{\pm0.0207}$ |
| GIN | $0.6131_{\pm0.0990}$ | $-0.5408_{\pm0.4193}$ | $0.9504_{\pm0.0007}$ |
| GCNII | $0.5287_{\pm0.0570}$ | $-1.1329_{\pm0.0135}$ | $0.7640_{\pm0.0355}$ |
| AMP | $-0.5891_{\pm0.0720}$ | $-3.9579_{\pm0.0769}$ | $0.0515_{\pm0.1819}$ |
| **DE-GNNs** | | | |
| DGC | $0.6028_{\pm0.0050}$ | $-0.1483_{\pm0.0231}$ | $0.8261_{\pm0.0032}$ |
| GRAND | $0.6715_{\pm0.0490}$ | $-0.0942_{\pm0.3897}$ | $0.6602_{\pm0.1393}$ |
| GraphCON | $0.0964_{\pm0.0620}$ | $-1.3836_{\pm0.0092}$ | $0.6833_{\pm0.0074}$ |
| A-DGN | $-0.5188_{\pm0.1812}$ | $-3.2417_{\pm0.0751}$ | $0.4296_{\pm0.1003}$ |
| SWAN | $-0.5981_{\pm0.1145}$ | $-3.5425_{\pm0.0830}$ | $-0.0739_{\pm0.2190}$ |
| PH-DGN | $-0.5473_{\pm0.1074}$ | $-4.2993_{\pm0.0721}$ | $-0.9348_{\pm0.2097}$ |
| **Graph Transformers** | | | |
| GPS | $-0.5121_{\pm0.0426}$ | $-3.5990_{\pm0.1949}$ | $0.6077_{\pm0.0282}$ |
| **Multi-hop GNNs** | | | |
| DRew-GCN | $-2.3692_{\pm0.1054}$ | $-1.5905_{\pm0.0034}$ | $-2.1004_{\pm0.0256}$ |
| + delay | $-2.4018_{\pm0.1097}$ | $-1.6023_{\pm0.0078}$ | $-2.0291_{\pm0.0240}$ |
| **Our** | | | |
| SONAR | $-3.2906_{\pm0.0706}$ | $-6.7517_{\pm0.0590}$ | $-3.1187_{\pm0.0192}$ |

**Results.** We report results using the $log_{10}(\text{MSE})$ metric in Table 1. This benchmark focuses on predicting shortest-path-based properties that inherently require the model to capture the global graph structure, thus long-range information. SONAR sets a new performance standard by outperforming all existing methods by at least one order of magnitude across all tasks. While standard MPNNs struggle to model such properties due to their limited ability to propagate information over long distances, SONAR consistently surpasses DE-GNNs (its direct competitors), multi-hop GNNs, and transformer-based models, which are typically more computationally demanding. Notably, SONAR improves over the best-performing baseline by 0.89 points on Diam., 1.01 on Ecc., and a remarkable 2.46 on the SSSP task, further demonstrating its effectiveness in capturing long-range dependencies.

## 4.3 Long-Range Graph Benchmark

**Setup.** To assess the performance on real-world long-range benchmarks, we consider the "Peptides-func", "Peptides-struct", "PascalVOC-SP" tasks [24]. The first two tasks involve predicting the functional or structural properties of graphs derived from peptides, which are large molecules based on amino acid chains. Similarly to previous tasks, accurate predictions require effectively capturing long-range interactions, as these properties are determined by the interplay between distant regions of the graphs, i.e., peptides. The "PascalVOC-SP" is a node-classification task on superpixel

graphs, which is considered to be more difficult than the peptides ones in the long-range context [6]. We use the same data and experimental setting in [24], including the 500k parameter budget.

**Results.** We present the performance of SONAR alongside leading baselines in Table 2, with a more extended comparison in Appendix C.1. SONAR achieves significantly better results than standard MPNNs and generally outperforms most DE-GNNs (i.e., SONAR's model class) and multi-hop GNNs across all tasks. On `func`, SONAR ranks second overall, only one point behind DRew, which relies on expensive graph rewiring. On `struct`, SONAR achieves a performance with standard deviation overlapping with top models. On `PascalVOC-SP`, SONAR improves the F1 score of transformer models by a remarkable 8.3%, ranking first with or without positional encoding. Overall, compared to transformer-based models with quadratic complexity, SONAR achieves comparable or better performance while maintaining linear complexity. We conclude that the long-range capabilities of SONAR are also evident in real-world tasks.

Table 2: Results for Peptides-func, Peptides-struct and PascalVOC-SP averaged over 3 training seeds. Baseline results are taken from [24, 46, 51, 65, 40, 52]. Note that all MPNN-based methods include structural and positional encoding. The **first**, **second**, and **third** best scores are colored.

| Model | Peptides-func
AP ↑ | Peptides-struct
MAE ↓ | Pascal VOC-SP
F1 ↑ |
|---|---|---|---|
| **MPNNs** | | | |
| GatedGCN | $58.64_{\pm 0.77}$ | $0.3420_{\pm 0.0013}$ | $0.2873_{\pm 0.0219}$ |
| GCN | $59.30_{\pm 0.23}$ | $0.3496_{\pm 0.0013}$ | $0.1268_{\pm 0.0060}$ |
| GCNII | $55.43_{\pm 0.78}$ | $0.3471_{\pm 0.0010}$ | $0.1698_{\pm 0.0080}$ |
| GINE | $54.98_{\pm 0.79}$ | $0.3547_{\pm 0.0045}$ | $0.1265_{\pm 0.0076}$ |
| **Multi-hop GNNs** | | | |
| DIGL+MPNN+LapPE | $68.30_{\pm 0.26}$ | $0.2616_{\pm 0.0018}$ | $0.2921_{\pm 0.0038}$ |
| DRew-GCN | $69.96_{\pm 0.76}$ | $0.2781_{\pm 0.0028}$ | $0.1848_{\pm 0.0107}$ |
| DRew-GCN+LapPE | $71.50_{\pm 0.44}$ | $0.2536_{\pm 0.0015}$ | $0.1851_{\pm 0.0092}$ |
| MixHop-GCN | $65.92_{\pm 0.36}$ | $0.2921_{\pm 0.0023}$ | $0.2506_{\pm 0.0133}$ |
| MixHop-GCN+LapPE | $68.43_{\pm 0.49}$ | $0.2614_{\pm 0.0023}$ | $0.2218_{\pm 0.0174}$ |
| **Transformers** | | | |
| GraphGPS+LapPE | $65.35_{\pm 0.41}$ | $0.2500_{\pm 0.0005}$ | **$0.3748_{\pm 0.0109}$** |
| Graph ViT | $69.42_{\pm 0.75}$ | $0.2449_{\pm 0.0016}$ | − |
| GRIT | $69.88_{\pm 0.82}$ | $0.2460_{\pm 0.0012}$ | − |
| SAN+LapPE | $63.84_{\pm 1.21}$ | $0.2683_{\pm 0.0043}$ | $0.3230_{\pm 0.0039}$ |
| Transformer+LapPE | $63.26_{\pm 1.26}$ | $0.2529_{\pm 0.0016}$ | $0.2694_{\pm 0.0098}$ |
| **DE-GNNs** | | | |
| GRAND | $57.89_{\pm 0.62}$ | $0.3418_{\pm 0.0015}$ | $0.1918_{\pm 0.0097}$ |
| GraphCON | $60.22_{\pm 0.68}$ | $0.2778_{\pm 0.0018}$ | $0.2108_{\pm 0.0091}$ |
| A-DGN | $59.75_{\pm 0.44}$ | $0.2874_{\pm 0.0021}$ | $0.2349_{\pm 0.0054}$ |
| SWAN | $67.51_{\pm 0.39}$ | $0.2485_{\pm 0.0009}$ | $0.3192_{\pm 0.0250}$ |
| PH-DGN | **$70.12_{\pm 0.45}$** | **$0.2465_{\pm 0.0020}$** | − |
| **Ours** | | | |
| SONAR | $68.42_{\pm 0.11}$ | $0.2525_{\pm 0.0038}$ | $0.4058_{\pm 0.0039}$ |
| SONAR+LapPE | $70.47_{\pm 0.41}$ | $0.2486_{\pm 0.0006}$ | $0.4082_{\pm 0.0037}$ |

## 4.4 Heterophilic Tasks

**Setup.** To further evaluate the performance of our SONAR, we assess its the effectiveness in capturing complex relational information in heterophilic settings, where nodes belonging to same class are often connected through longer and sparser paths, we consider the five node classification tasks introduced in [70]. Specifically, we consider the "Roman-empire", "Amazon-ratings", "Minesweeper", "Tolokers", and "Questions" datasets. We adhere to the same data and experimental setting presented in [70].

**Results.** We report the results in Table 3 (extended comparison in Appendix C.1). SONAR achieves the best performance on both the `Roman-empire` and `Minesweeper` tasks, surpassing the second-best models by 0.8 and 2.8 points, respectively. It also performs competitively on `Amazon-ratings` and `Tolokers`, with less than one point difference from the top models and overlapping stds. Remarkably, SONAR outperforms all heterophily-designated GNNs by up to 20 points, except for the `Questions` dataset. This shows the flexibility of our approach on different tasks.

## 4.5 Empirical Sensitivity Analysis

We empirically assess the long-range capabilities of SONAR compared to a standard GCN, which can be considered as the most comparable baseline to our proposed SONAR. Both rely on the Laplacian operator and message-passing paradigm, but GCN lacks energy preservation guarantees for long-range information propagation, adaptive resistance, and external forces. For this analysis, we consider the Line task in Section 4.1 with $\ell = 50$ and measure the norm of the sensitivity matrix in Equation (13) between any node $v$ and the source node $u$. Specifically, we compute the sensitivity considering increasing distance between $u$ and $v$ (i.e., 10, 20, 30, 40, 50) with respect to increasing values of model recurrences (i.e., number of explored hops). The results, reported in Table 4, confirm our theoretical findings in Section 3.3: the information propagation of SONAR leads to sensitivity norms that never vanish, even at higher distances. On the contrary, the dissipative dynamics of the GCN cause an exponential decay in the influence from the source node to distant nodes.

Table 3: Mean test set score and std averaged over 4 random weight initializations on heterophilic datasets. The higher, the better. First, second, and **third** best results for each task are color-coded.

| Model | Roman-empire Acc ↑ | Amazon-ratings Acc ↑ | Minesweeper AUC ↑ | Tolokers AUC ↑ | Questions AUC ↑ |
|---|---|---|---|---|---|
| **MPNNs** | | | | | |
| GAT | $80.87_{\pm 0.30}$ | $49.09_{\pm 0.63}$ | $92.01_{\pm 0.68}$ | $83.70_{\pm 0.47}$ | $77.43_{\pm 1.20}$ |
| Gated-GCN | $74.46_{\pm 0.54}$ | $43.00_{\pm 0.32}$ | $87.54_{\pm 1.22}$ | $77.31_{\pm 1.14}$ | $76.61_{\pm 1.13}$ |
| GCN | $73.69_{\pm 0.74}$ | $48.70_{\pm 0.63}$ | $89.75_{\pm 0.52}$ | $83.64_{\pm 0.67}$ | $76.09_{\pm 1.27}$ |
| SAGE | $85.74_{\pm 0.67}$ | $53.63_{\pm 0.39}$ | $93.51_{\pm 0.57}$ | $82.43_{\pm 0.44}$ | $76.44_{\pm 0.62}$ |
| **Graph Transformers** | | | | | |
| Exphormer | $89.03_{\pm 0.37}$ | $53.51_{\pm 0.46}$ | $90.74_{\pm 0.53}$ | $83.77_{\pm 0.78}$ | $73.94_{\pm 1.06}$ |
| NAGphormer | $74.34_{\pm 0.77}$ | $51.26_{\pm 0.72}$ | $84.19_{\pm 0.66}$ | $78.32_{\pm 0.95}$ | $68.17_{\pm 1.53}$ |
| GOAT | $71.59_{\pm 1.25}$ | $44.61_{\pm 0.50}$ | $81.09_{\pm 1.02}$ | $83.11_{\pm 1.04}$ | $75.76_{\pm 1.66}$ |
| GPS | $82.00_{\pm 0.61}$ | $53.10_{\pm 0.42}$ | $90.63_{\pm 0.67}$ | $83.71_{\pm 0.48}$ | $71.73_{\pm 1.47}$ |
| GPS$_{\text{GAT+Performer}}$ (RWSE) | $87.04_{\pm 0.58}$ | $49.92_{\pm 0.68}$ | $91.08_{\pm 0.58}$ | $84.38_{\pm 0.91}$ | $77.14_{\pm 1.49}$ |
| GT | $86.51_{\pm 0.73}$ | $51.17_{\pm 0.66}$ | $91.85_{\pm 0.76}$ | $83.23_{\pm 0.64}$ | $77.95_{\pm 0.68}$ |
| GT-sep | $87.32_{\pm 0.39}$ | $52.18_{\pm 0.80}$ | $92.29_{\pm 0.47}$ | $82.52_{\pm 0.92}$ | $78.05_{\pm 0.93}$ |
| **Heterophily-Designated GNNs** | | | | | |
| FAGCN | $65.22_{\pm 0.56}$ | $44.12_{\pm 0.30}$ | $88.17_{\pm 0.73}$ | $77.75_{\pm 1.05}$ | $77.24_{\pm 1.26}$ |
| FSGNN | $79.92_{\pm 0.56}$ | $52.74_{\pm 0.83}$ | $90.08_{\pm 0.70}$ | $82.76_{\pm 0.61}$ | $78.86_{\pm 0.92}$ |
| GBK-GNN | $74.57_{\pm 0.47}$ | $45.98_{\pm 0.71}$ | $90.85_{\pm 0.58}$ | $81.01_{\pm 0.67}$ | $74.47_{\pm 0.86}$ |
| GPR-GNN | $64.85_{\pm 0.27}$ | $44.88_{\pm 0.34}$ | $86.24_{\pm 0.61}$ | $72.94_{\pm 0.97}$ | $55.48_{\pm 0.91}$ |
| JacobiConv | $71.14_{\pm 0.42}$ | $43.55_{\pm 0.48}$ | $89.66_{\pm 0.40}$ | $68.66_{\pm 0.65}$ | $73.88_{\pm 1.16}$ |
| **Our** | | | | | |
| SONAR | $89.82_{\pm 0.57}$ | $52.22_{\pm 0.14}$ | $96.29_{\pm 0.73}$ | $83.57_{\pm 1.44}$ | $74.96_{\pm 1.10}$ |

Table 4: Sensitivity across different distances and recurrences in the Line-50 graph from Section 4.1.

| | SONAR | | | | | GCN | | | |
|---|---|---|---|---|---|---|---|---|---|
| N. Recurrences→ Distance ↓ | 25 | 50 | 75 | 100 | N. Recurrences→ Distance ↓ | 25 $\times 10^{-5}$ | 50 $\times 10^{-5}$ | 75 $\times 10^{-5}$ | 100 $\times 10^{-5}$ |
| **10** | 0.0115 | 0.0413 | 0.573 | 1.955 | **10** | 0.7078 | 0.0 | 1.1470 | 0.3562 |
| **20** | 0.0037 | 0.0334 | 0.703 | 5.084 | **20** | 0.0 | 0.0 | 1.0490 | 0.2980 |
| **30** | 0.0273 | 0.0616 | 1.269 | 4.341 | **30** | 0.3725 | 0.9779 | 0.5259 | 0.0745 |
| **40** | 0.0131 | 0.0253 | 0.584 | 1.061 | **40** | 0.0 | 0.0 | 0.0 | 0.0 |
| **50** | 0.0002 | 0.0324 | 0.143 | 1.004 | **50** | 0.0 | 0.0 | 0.0 | 0.0 |

## 5 Conclusions

In this work, we introduced SONAR a novel DE-GNN architecture that models information propagation using wave dynamics governed by the graph wave equation. SONAR offers a principled approach for long-range information propagation by balancing conservative and non-conservative behaviors by integrating adaptive edge resistances and state-dependent external forces. Our theoretical analysis demonstrates that SONAR's energy preservation ensures stable and effective propagation of information over long distances. Moreover, the sensitivity analysis confirms that SONAR maintains non-vanishing influence between distant nodes, both in its continuous and discretized forms. Empirically, SONAR achieves state-of-the-art performance on a range of challenging benchmarks, i.e., synthetic and real-world long-range tasks, as well as heterophilic tasks. For such a reason, we believe SONAR represents a step forward in the design of GNNs capable of effective long-range propagation. Future work can focus on exploring alternative discretization methods, e.g., adaptive multistep scheme [4], and extend SONAR to temporal graphs [38].

**Impact Statement.** This work aims to advance the field of machine learning on graphs, with a focus on enhancing long-range information propagation. There are many potential societal consequences of our work, none which we feel must be specifically highlighted here.

## Acknowledgments and Disclosure of Funding

The work has been partially supported by EU-EIC EMERGE (Grant No. 101070918).

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

# A Experimental Details

## A.1 Edge Feature Aggregation

Together with the usual node features $\bar{\mathbf{x}}_v \in \mathbb{R}^d$, some graphs provide edge features $\mathbf{E} \in \mathbb{R}^{m \times d_e}$ made of a $d_e$-dimensional vector $\mathbf{e}_{uv} \in \mathbb{R}^{d_e}$ for every edge $(u, v) \in \mathcal{E}$. While classical MPNNs add this information during the convolution, we adopt a different approach to preserve our wave-like propagation and theoretical properties. In particular, we follow [26], transforming the edge features and injecting them into the node space. To do so, we define the average Avg and gradient Grad operators, which shift node features into edge features [26, 33], as

$$(\text{Avg}\mathbf{X})_{uv} = \frac{1}{2}\mathbf{G}_{uv}(\mathbf{x}_v + \mathbf{x}_u), \qquad (\text{Grad}\mathbf{X})_{uv} = \mathbf{G}_{uv}(\mathbf{x}_v - \mathbf{x}_u), \qquad (14)$$

where $\mathbf{G} \in \mathbb{R}^{n \times n}$ is a matrix where $\mathbf{G}_{uv} = \frac{1}{\gamma_{uv}}$ and $\gamma_{uv}$ is the geometric mean of the degrees of nodes $u$ and $v$. Their adjoint operators, which coincide with their transpose, transform edge features into node features [33, 26]. Therefore, before starting the propagation of information, the initial condition is a concatenation of the initial node features and the transformation of the edge ones:

$$\bar{\mathbf{X}}_{\text{new}} = \left(\bar{\mathbf{X}} \oplus \text{Avg}^\top \mathbf{E} \oplus \text{Grad}^\top \mathbf{E}\right) \qquad (15)$$

## A.2 Employed Baselines

In our experiments, the performance of our method is compared with various state-of-the-art GNN baselines from the literature. Specifically, we consider:

- classical MPNN-based methods, i.e., GCN [59], GraphSAGE [49], GAT [85], GatedGCN [9], GIN [89], GINE [54], GCNII [15];
- heterophily-specific models, i.e., H2GCN [93], CPGNN [92], FAGCN [8], GPR-GNN [17], FSGNN [66], GloGNN [63], GBK-GNN [22], and JacobiConv [86];
- DE-GNNs, i.e., DGC [87], GRAND [13], GraphCON [75], A-DGN [39], SWAN [40], PH-DGN [52];
- Graph Transformers, i.e., Transformer [84, 23], GT [79], SAN [61], GPS [72], GOAT [60], Exphormer [80], NAGphormer [14], GRIT[65], and GraphViT [51];
- Higher-Order GNNs, i.e., DIGL [35], MixHop [1], DRew [46], and GRED [20]. .

## A.3 Employed Datasets

In our experiments, we evaluate SONAR's performance on widely used graph benchmarks. Specifically, we consider long-range propagation tasks, including the graph transfer tasks from [40], as well as the three graph property prediction tasks introduced in [39] ("Diameter", "SSSP", and "Eccentricity"). Additionally, we assess SONAR on the "Peptide-func", "Peptide-struct", and "PascalVOC-SP" tasks from the real-world Long-Range Graph Benchmark (LRGB) [24]. To further evaluate its effectiveness, we include five heterophilic tasks: "Roman-empire", "Amazon-ratings", "Minesweeper", "Tolokers", and "Questions" [70]. We highlight that the three graph property prediction tasks are problems fundamentally linked with information propagation, as node signals must travel from each node $v$ across the entire graph. Therefore, they require nodes to iteratively exchange information with increasingly distant nodes. This process shares similarities with classical algorithms like Bellman-Ford or Dijkstra's algorithm. Indeed, Dijkstra's algorithm works by progressively expanding a frontier of visited nodes, where each node updates the shortest known distance to its neighbors.

In Table 5, we report the statistics of the employed datasets.

## A.4 Hyperparameter Space

The hyperparameter space employed by SONAR in our experiments is reported in Table 6.

Table 5: Dataset statistics.

| Dataset | #Nodes | #Edges | #Graphs | Task |
|---|---|---|---|---|
| Graph Transfer | 3 - 100 | 7 - 396 | 1,200 | Node Regression |
| Diameter | 25 - 35 | 22 - 553 | 7,040 | Graph Regression |
| SSSP | 25 - 35 | 22 - 553 | 7,040 | Node Regression |
| Eccentricity | 25 - 35 | 22 - 553 | 7,040 | Node Regression |
| Peptide-func | 8 - 444 | 10 - 928 | 15,535 | Graph Classification |
| Peptide-struct | 8 - 444 | 10 - 928 | 15,535 | Graph Regression |
| PascalVOC-SP | 198 - 500 | 1,044 - 2,942 | 11,355 | Node Classification |
| Roman-empire | 22,662 | 32,927 | 1 | Node Classification |
| Amazon-ratings | 24,492 | 93,050 | 1 | Node Classification |
| Minesweeper | 10,000 | 39,402 | 1 | Node Classification |
| Tolokers | 11,758 | 519,000 | 1 | Node Classification |
| Questions | 48,921 | 153,540 | 1 | Node Classification |

Table 6: The grid of hyperparameters employed during model selection for the graph transfer tasks (*Transfer*), graph property prediction tasks (*GPP*), Long Range Graph Benchmark (*LRGB*), and heterophilic benchmarks (*Hetero*).

| Hyperparameters | Values | | | |
|---|---|---|---|---|
| | *Transfer* | *GPP* | *LRGB* | *Hetero* |
| Optimizer | Adam | Adam | AdamW | AdamW |
| Learning rate | 0.001 | 0.003 | 0.001 | 0.001, 0.0005 |
| Weight decay | 0 | $10^{-6}$ | 0 | 0 |
| Dropout | 0 | 0 | 0, 0.2, 0.5 | 0, 0.2, 0.5 |
| N. recurrences | distance/2, distance | 5, 10, 20 | 1,2,4,6,8,12,16 | 1,2,4,6,8,12,16 |
| Embedding dim | 64 | 30 | 60, 68, 74, 105, 117, 136 | 64, 128, 256, 512 |
| N. Blocks | 1,2 | 1, 2 | 1,2,3,4,5 | from 1 to 12 |
| $\epsilon$ | 0.05, 0.1, 0.5, 1 | 0.001, 0.1, 0.5, 1 | 0.01, 0.02, 0.025, 0.05, 0.1 | 0.0001, 0.0005, 0.001, 0.005, 0.01, 0.05, 0.1 |
| Use Dissipation | | | True, False | |
| Use External Force | | | True, False | |
| Use Adaptive Resistance | | | True, False | |
| $\mathbf{L}$ | | | $\mathbf{D} - \mathbf{A}, \quad \mathbf{I} - \mathbf{D}^{-1/2}\mathbf{A}\mathbf{D}^{-1/2}, \quad \mathbf{I} - \mathbf{D}^{-1}\mathbf{A}$ | |

# B  Proofs of the Theoretical Results

In this section, we prove the theoretical results in Section 3.3.

## B.1  Proof of Theorem 3.1

To prove Theorem 3.1, we first need this preliminary lemma:

**Lemma B.1.** *Let* $\mathbf{X} \in \mathbb{R}^n$ *be a feature vector for the nodes in graph* $\mathcal{G}$*. Then, it holds that*

$$\sum_{v \in V} \|\nabla \mathbf{x}_v(t)\|^2 = \sum_{v \in V} \sum_{u \in \mathcal{N}_v} (\mathbf{x}_v - \mathbf{x}_u)^2 = \frac{1}{2} \sum_{u,v \in V} \mathbf{A}_{vu}(\mathbf{x}_v - \mathbf{x}_u)^2 = \mathbf{X}^\top \mathbf{L} \mathbf{X} \qquad (16)$$

The lemma involves only one feature per node, but it can easily be generalized to multiple features by considering norms and scalar products.

*Proof of Lemma B.1.* The first equivalence comes from the gradient definition in a graph (see [34]), while the second follows from expanding the sum of neighbors. We now show the last equivalence:

$$
\begin{aligned}
\frac{1}{2} \sum_{u,v \in V} \mathbf{A}_{vu} (\mathbf{x}_v - \mathbf{x}_u)^2 &= \frac{1}{2} \sum_v \sum_u \mathbf{A}_{uv} (\mathbf{x}_v^2 + \mathbf{x}_u^2 - 2\mathbf{x}_v \mathbf{x}_u) \\
&= \frac{1}{2} \sum_v \mathbf{x}_v^2 \sum_u \mathbf{A}_{uv} + \sum_u \mathbf{x}_u^2 \sum_v \mathbf{A}_{vu} - 2 \sum_{u,v} \mathbf{A}_{vu} \mathbf{x}_v \mathbf{x}_u \\
&= \frac{1}{2} \left( \sum_v \mathbf{x}_v^2 \deg(v) + \sum_u \mathbf{x}_u^2 \deg(u) - 2 \sum_{uv} \mathbf{A}_{uv} \mathbf{x}_u \mathbf{x}_v \right) \\
&= \frac{1}{2} \left( 2 \sum_v \mathbf{x}_v^2 \mathbf{D}_{vv} - 2 \sum_{uv} \mathbf{A}_{uv} \mathbf{x}_u \mathbf{x}_v \right) \\
&= \sum_{v \in V} \mathbf{x}_v \left( \mathbf{x}_v \mathbf{D}_{vv} - \sum_u \mathbf{A}_{vu} \mathbf{x}_u \right) \\
&= \sum_{v \in V} \mathbf{x}_v \left( \mathbf{x}_v \mathbf{D}_{vv} \sum_u (\delta_{uv} \mathbf{x}_u - \mathbf{A}_{vu} \mathbf{x}_u) \right) \\
&= \sum_{v \in V} \mathbf{x}_v \sum_u (\mathbf{D}_{uu} \delta_{uv} - \mathbf{A}_{vu}) \mathbf{x}_u \\
&= \sum_v \sum_u \mathbf{x}_v (\mathbf{D}_{uu} \delta_{uv} - \mathbf{A}_{uv}) \mathbf{x}_u \\
&= \sum_{uv} \mathbf{x}_u \mathbf{L} \mathbf{x}_v = \mathbf{X}^\top \mathbf{L} \mathbf{X}
\end{aligned}
\tag{17}
$$

where $\delta_{uv}$ is 1 if $u = v$ and 0 otherwise. $\qquad\square$

*Proof of Theorem 3.1.* We show the theorem for $d = 1$ (one feature), since it involves the norms of the feature vector $\mathbf{X}(t)$. To improve clarity, here we consider the initial considtion to be $\mathbf{X}(0) = \bar{\mathbf{X}} = \bar{\mathbf{X}}$. The wave (4) has an explicit solution (see [34]) given by

$$
\mathbf{X}(t) = \cos\left( t \sqrt{\mathbf{L}} \right) \bar{\mathbf{X}},
\tag{18}
$$

where

$$
\cos\left( t \sqrt{\mathbf{L}} \right) = I - \frac{t^2}{2} \mathbf{L} + \frac{t^4}{4!} \mathbf{L}^2 + \ldots = \sum_{n=0}^{\infty} (-1)^n \frac{t^{2n}}{n!} \sqrt{\mathbf{L}}^{2n}.
\tag{19}
$$

The energy in (9) can be calculated as

$$
\begin{aligned}
E(t) &= \frac{1}{2} \left( \sum_{v \in V} \|\nabla \mathbf{x}_v(t)\|^2 + \left\| \frac{\partial \mathbf{X}(t)}{\partial t} \right\|^2 \right) \\
&= \frac{1}{2} \left( \mathbf{X}(t)^\top \mathbf{L} \mathbf{X}(t) + \left( \frac{\partial \left( \cos\left( t \sqrt{\mathbf{L}} \right) \bar{\mathbf{X}} \right)}{\partial t} \right)^2 \right),
\end{aligned}
\tag{20}
$$

where we used the Lemma B.1 for the first term and the explicit solution for the second. Using the explicit solution, the first term is equal to

$$
\begin{aligned}
\mathbf{X}(t)^\top \mathbf{L} \mathbf{X}(t) &= \left( \cos\left( t \sqrt{\mathbf{L}} \right) \bar{\mathbf{X}} \right)^\top \mathbf{L} \left( \cos\left( t \sqrt{\mathbf{L}} \right) \bar{\mathbf{X}} \right) \\
&= (\bar{\mathbf{X}})^\top \cos\left( t \sqrt{\mathbf{L}} \right)^\top \mathbf{L} \cos\left( t \sqrt{\mathbf{L}} \right) \bar{\mathbf{X}} \\
&= \left\| \sqrt{\mathbf{L}} \cos\left( t \sqrt{\mathbf{L}} \right) \bar{\mathbf{X}} \right\|^2 \\
&= \left\| \cos\left( t \sqrt{\mathbf{L}} \right) \sqrt{\mathbf{L}} \bar{\mathbf{X}} \right\|^2,
\end{aligned}
\tag{21}
$$

where the last equality follows from the fact that $\cos\left(t\sqrt{\mathbf{L}}\right)$ and $\sqrt{\mathbf{L}}$ commute, which follows from the Taylor series expansion in (19). The second term can be calculated as

$$\frac{\partial}{\partial t}\left(\sum_{n=0}^{\infty}(-1)^n\frac{t^{2n}}{2n!}\sqrt{\mathbf{L}}^{2n}\right)\bar{\mathbf{X}} = \left(\sum_{n=1}^{\infty}(-1)^{n-1}\frac{t^{2n-1}}{(2n-1)!}\sqrt{\mathbf{L}}^{2n}\right)\bar{\mathbf{X}}. \qquad (22)$$

which, using the Taylor series expansion of $\sin\left(t\sqrt{\mathbf{L}}\right)$, is equal to

$$\left\|\frac{\partial\mathbf{X}(t)}{\partial t}\right\|^2 = \left\|\sqrt{\mathbf{L}}\sin\left(t\sqrt{\mathbf{L}}\right)\bar{\mathbf{X}}\right\|^2 = \left\|\sin\left(t\sqrt{\mathbf{L}}\right)\sqrt{\mathbf{L}}\bar{\mathbf{X}}\right\|^2 \qquad (23)$$

Finally, the sum of the two terms is equal to

$$\begin{aligned}
E(t) &= \frac{1}{2}\left(\left\|\cos\left(t\sqrt{\mathbf{L}}\right)\sqrt{\mathbf{L}}\bar{\mathbf{X}}\right\|^2 + \left\|\sin\left(t\sqrt{\mathbf{L}}\right)\sqrt{\mathbf{L}}\bar{\mathbf{X}}\right\|^2\right) \\
&= \frac{1}{2}\left((\bar{\mathbf{X}})^{\top}\sqrt{\mathbf{L}}^{\top}\cos\left(t\sqrt{\mathbf{L}}\right)^{\top}\cos\left(t\sqrt{\mathbf{L}}\right)\sqrt{\mathbf{L}}\bar{\mathbf{X}} + (\bar{\mathbf{X}})^{\top}\sqrt{\mathbf{L}}^{\top}\sin\left(t\sqrt{\mathbf{L}}\right)^{\top}\sin\left(t\sqrt{\mathbf{L}}\right)\sqrt{\mathbf{L}}\bar{\mathbf{X}}\right) \\
&= \frac{1}{2}\left((\bar{\mathbf{X}})^{\top}\sqrt{\mathbf{L}}\cos^2\left(t\sqrt{\mathbf{L}}\right)\sqrt{\mathbf{L}}\bar{\mathbf{X}} + (\bar{\mathbf{X}})^{\top}\sqrt{\mathbf{L}}\sin^2\left(t\sqrt{\mathbf{L}}\right)\sqrt{\mathbf{L}}\bar{\mathbf{X}}\right) \\
&= \frac{1}{2}\left((\bar{\mathbf{X}})^{\top}\sqrt{\mathbf{L}}\left(\cos^2\left(t\sqrt{\mathbf{L}}\right) + \sin^2\left(t\sqrt{\mathbf{L}}\right)\right)\sqrt{\mathbf{L}}\bar{\mathbf{X}}\right) \\
&= \frac{1}{2}\left((\bar{\mathbf{X}})^{\top}\sqrt{\mathbf{L}}\sqrt{\mathbf{L}}\bar{\mathbf{X}}\right) = \frac{1}{2}\left((\bar{\mathbf{X}})^{\top}\mathbf{L}\bar{\mathbf{X}}\right) = E(0),
\end{aligned}$$
$$(24)$$

Where the transpositions for $\sqrt{\mathbf{L}}$, $\sin$, and $\cos$ vanish as they are all symmetric matrices (since $\mathbf{L}$ is symmetric and positive definite, we can choose a symmetric square root). $\qquad\square$

We now prove a general version of Theorem 3.1 with $d$ features $\mathbf{X}(t) \in \mathbb{R}^{n\times d}$ and $\mathbf{W} = \mathbf{I}$.

**Theorem B.1.** *Let $\mathbf{X}(t) \in \mathbb{R}^{n\times d}$ be the node states at time $t$, obtained as the solution to the graph wave equation in Equation (5), with initial condition $\mathbf{X}(0) = \bar{\mathbf{X}}$, null dissipative and external forcing terms, and $\mathbf{W} = \mathbf{I}$. Then, the energy $E(t)$, defined as*

$$E(t) = \sum_{v\in V}\frac{1}{2}\|\nabla\mathbf{x}_v(t)\|^2 + \frac{1}{2}\left\|\frac{\partial\mathbf{X}(t)}{\partial t}\right\|^2, \qquad (25)$$

*is conserved, that is, $E(t) = E(0)\ \forall t > 0$.*

*Proof.* The proof is almost identical to the one from Theorem Theorem 3.1. Since features are not coupled, the solution to the equation is the same as Equation (10), and the same steps apply. Since we work with multiple features, scalar norms become vector norms, while the latter become matrix norms. $\qquad\square$

## B.2 Proof of Theorem 3.2

While the proof of Theorem 3.2 is a direct consequence of the explicit solution, we now prove a more general version of it for node states with $d$ features and $\mathbf{W} = \mathbf{I}$.

**Theorem B.2** (Sensitivity matrix, continuous case). *Let $\mathbf{x}_v(t) \in \mathbb{R}^d$ be the state for node $v$ at time $t$. Then, the sensitivity matrix $\frac{\partial\mathbf{x}_v(t)}{\partial\mathbf{x}_u(0)}$ between nodes $u$ and $v$ is*

$$\frac{\partial\mathbf{x}_v(t)}{\partial\mathbf{x}_u(0)} = \cos\left(t\sqrt{\mathbf{L}}\right)_{vu}\mathbf{I}. \qquad (26)$$

*Proof.* Since $\mathbf{W} = \mathbf{I}$, the analytic solution to the SONAR wave equation in Equation (4) is the vector version of Equation (10)

$$\mathbf{X}(t) = \cos\left(t\sqrt{\mathbf{L}}\right)\bar{\mathbf{X}} = \cos\left(t\sqrt{\mathbf{L}}\right)\mathbf{X}(0) \qquad (27)$$

Now, by differentiating with respect to $\mathbf{x}_u(0)$, we obtain

$$\frac{\partial \mathbf{x}_v(t)}{\partial \mathbf{x}_u(0)} = \cos\left(t\sqrt{\mathbf{L}}\right)_{vu} \mathbf{I}. \tag{28}$$

$\square$

While Theorem B.1 and Theorem B.2 are not the most general statements with any possible feature coupling, we are the first to show proofs for the wave equation for graphs involving multidimensional features. Most of the works in the mathematical literature on the graph-based Laplacian and graph wave equation involve only the scalar case with single-valued functions [34, 53]

## B.3  Proof of Theorem 3.3

*Proof.* We start by recalling the Sonar update with null dissipative and external forcing term

$$\begin{cases} \mathbf{X}^{\ell+1} = \mathbf{X}^\ell + h\mathbf{V}^{\ell+1}, \\ \mathbf{V}^{\ell+1} = \mathbf{V}^\ell - h\left(\mathbf{L}\mathbf{X}^\ell\mathbf{W}\right), \end{cases} \tag{29}$$

Substituting the second equation in the first one, we obtain

$$\begin{aligned} \mathbf{X}^{\ell+1} &= \mathbf{X}^\ell + h\mathbf{V}^\ell - h^2\mathbf{L}\mathbf{X}^\ell\mathbf{W} \\ &= \mathbf{X}^\ell + h\frac{\mathbf{X}^\ell - \mathbf{X}^{\ell-1}}{h} - h^2\mathbf{L}\mathbf{X}^\ell\mathbf{W} \\ &= 2\mathbf{X}^\ell - \mathbf{X}^{\ell-1} - h^2\mathbf{L}\mathbf{X}^\ell\mathbf{W} \end{aligned} \tag{30}$$

We now write the update for a single node $v$

$$\mathbf{x}_v^{\ell+1} = 2\mathbf{x}_v^\ell - \mathbf{x}_v^{\ell-1} - h^2\sum_{u\in\mathcal{V}}\mathbf{L}_{vu}\mathbf{x}_u^\ell\mathbf{W} \tag{31}$$

Therefore, if we consider any node $u \in \mathcal{V}$, the one-step sensitivity matrix can be directly calculated by differentiating Equation 31 by $\mathbf{x}_u$. Since the dynamical system is causal, we have that $\frac{\partial \mathbf{x}_v^{\ell-1}}{\partial \mathbf{x}_u^\ell} = 0$ and, finally,

$$\frac{\partial \mathbf{x}_v^\ell}{\partial \mathbf{x}_u^{\ell-1}} = 2\mathbf{I}_{uv} - h^2\mathbf{L}_{uv}\mathbf{W}, \tag{32}$$

with $\mathbf{I}_{uv}$ denoting the $(u, v)$-th entry of the identity matrix. $\square$

## B.4  Proof of Theorem 3.4

*Proof of Theorem 3.4.* Let us consider one update of the wave equation solution from equation 7

$$\begin{aligned} \mathbf{X}^{\ell+1} &= \mathbf{X}^\ell + h\mathbf{V}^{\ell+1} \\ \mathbf{V}^{\ell+1} &= \mathbf{V}^\ell - hL\mathbf{X}^\ell\mathbf{W} - h\mathbf{k}(\mathbf{X}^\ell)\odot\mathbf{V}^\ell + h\mathbf{F}(\mathbf{X}^\ell) \end{aligned} \tag{33}$$

We will now show that

$$\left\|\frac{\partial \mathbf{x}_v^\ell}{\partial \mathbf{x}_u^0}\right\|_{L^1} \leq (wd)^\ell\left(\left((1 + h + h^2(N + k + 1))\mathbf{I} + h^2\mathbf{A}\right)^\ell\right)_{vu} \tag{34}$$

and

$$\left\|\frac{\partial \mathbf{v}_v^\ell}{\partial \mathbf{x}_u^0}\right\|_{L^1} \leq (wd)^\ell\left(\left((1 + h(N + k + 1))\mathbf{I} + h\mathbf{A}\right)^\ell\right)_{vu} \tag{35}$$

by induction, following [19].

**Base case $\ell = 1$** . Using the update equations, it is clear that

$$\left|\frac{\partial \mathbf{x}_v^{1,\alpha}}{\partial \mathbf{x}_u^{0,\beta}}\right| \leq \delta_{vu} + \delta_{vu}h\left|\frac{\partial \mathbf{v}_v^{1,\alpha}}{\partial \mathbf{x}_u^{0,\beta}}\right| \tag{36}$$

and

$$\left|\frac{\partial \mathbf{v}_v^{1,\alpha}}{\partial \mathbf{x}_u^{0,\beta}}\right| = \delta_{vu}\left(\left|\frac{\partial \mathbf{v}_v^{0,\alpha}}{\partial \mathbf{x}_u^{0,\beta}}\right| + hk\left|\frac{\partial \mathbf{v}_v^{0,\gamma}}{\partial \mathbf{x}_u^{0,\beta}}\right| + |h\mathbf{W}_F^{\alpha\beta}|\right) + h|\mathbf{L}_{vw}|\left|\frac{\partial \mathbf{x}_w^{0,\gamma}}{\partial \mathbf{x}_u^{0,\beta}}\right||\mathbf{W}^{\gamma\beta}|. \tag{37}$$

Since $\left|\frac{\partial \mathbf{v}_v^{0,\alpha}}{\partial \mathbf{x}_v^{0,\beta}}\right| = |\mathbf{W}_V^{\alpha\beta}| \leq w$, we have that

$$\left|\frac{\partial \mathbf{v}_v^{1,\alpha}}{\partial \mathbf{x}_u^{0,\beta}}\right| \leq \delta_{vu}w(1 + hk + h) + |\mathbf{L}_{vu}|wh \leq \delta_{vu}w\left(1 + h(N + k + 1)\right) + \mathbf{A}_{uv}wh \tag{38}$$

which, by summing over $\beta$ and maximizing on $\alpha$

$$\left\|\frac{\partial \mathbf{v}_v^1}{\partial \mathbf{x}_u^0}\right\|_{L^1} \leq (d)\left((1 + h(N + k + 1))\mathbf{I} + h\mathbf{A}\right)_{vu}, \tag{39}$$

and, by applying the same reasoning to equation 36

$$\left\|\frac{\partial \mathbf{x}_v^1}{\partial \mathbf{x}_u^0}\right\|_{L^1} \leq (wd)\left((1 + h + h^2(k + N + 1))\mathbf{I} + h^2\mathbf{A}\right)_{vu}. \tag{40}$$

**Inductive step, $\ell \to \ell + 1$.** We start by noticing that

$$\left|\frac{\partial \mathbf{x}_v^{\ell+1,\alpha}}{\partial \mathbf{x}_u^{0,\beta}}\right| \leq \delta_{vu}\left(\left|\frac{\partial \mathbf{x}_v^{\ell,\alpha}}{\partial \mathbf{x}_u^{0,\beta}}\right| + h\left|\frac{\partial \mathbf{v}_v^{\ell+1,\alpha}}{\partial \mathbf{x}_u^{0,\beta}}\right|\right), \tag{41}$$

which can be further decomposed as

$$\left|\frac{\partial \mathbf{x}_v^{\ell+1,\alpha}}{\partial \mathbf{x}_u^{0,\beta}}\right| \leq \left(\left|\frac{\partial \mathbf{x}_v^{\ell,\alpha}}{\partial \mathbf{x}_u^{0,\beta}}\right| + h\left|\frac{\partial \mathbf{v}_v^{\ell+1,\alpha}}{\partial \mathbf{x}_w^{\ell,\gamma}}\right|\left|\frac{\partial \mathbf{x}_w^{\ell,\gamma}}{\partial \mathbf{x}_u^{0,\beta}}\right| + h\left|\frac{\partial \mathbf{v}_v^{\ell+1,\alpha}}{\partial \mathbf{v}_w^{\ell,\gamma}}\right|\left|\frac{\partial \mathbf{v}_w^{\ell,\gamma}}{\partial \mathbf{x}_u^{0,\beta}}\right|\right). \tag{42}$$

It is easy to see that

$$\left|\frac{\partial \mathbf{v}_v^{\ell+1,\alpha}}{\partial \mathbf{x}_w^{\ell,\gamma}}\right| \leq h|\mathbf{L}_{vw}|w + hw\delta_{vu} \leq \delta_{vw}hw(N + 1) + wh\mathbf{A}_{vw},$$
$$\left|\frac{\partial \mathbf{v}_v^{\ell+1,\alpha}}{\partial \mathbf{v}_w^{\ell,\gamma}}\right| \leq \delta_{vw}(1 + hk). \tag{43}$$

We now calculate

$$\left|\frac{\partial \mathbf{v}_w^{\ell+1,\alpha}}{\partial \mathbf{x}_u^{0,\beta}}\right| \leq \delta_{wu}\left((1 + hk)\left|\frac{\partial \mathbf{v}_w^{\ell,\alpha}}{\partial \mathbf{x}_u^{0,\beta}}\right| + wh\left|\frac{\partial \mathbf{x}_w^{\ell,\alpha}}{\partial \mathbf{x}_u^{0,\beta}}\right|\right) + hw|\mathbf{L}_{wu}|\left|\frac{\partial \mathbf{x}_w^{\ell,\alpha}}{\partial \mathbf{x}_u^{0,\beta}}\right|$$
$$\leq \delta_{wu}\left((1 + hk)\left\|\frac{\partial \mathbf{v}_w^\ell}{\partial \mathbf{x}_u^0}\right\|_{L^1} + wh(N + 1)\left\|\frac{\partial \mathbf{x}_w^\ell}{\partial \mathbf{x}_u^0}\right\|_{L^1}\right) + hw\mathbf{A}_{wu}\left\|\frac{\partial \mathbf{x}_w^\ell}{\partial \mathbf{x}_u^0}\right\|_{L^1} \tag{44}$$

We now expand this last expression by inductive hypothesis and discard the terms with $O(h^2)$ as they become $O(h^3)$ in the next step. Summing over $\beta$ and maximizing over $\alpha$ we get

$$\left|\frac{\partial \mathbf{v}_w^{\ell+1,\alpha}}{\partial \mathbf{x}_u^{0,\beta}}\right| \leq (wd)^{\ell+1}\left(((1 + h(N + k + 1))\mathbf{I} + h\mathbf{A})^{\ell+1}\right)_{vu}. \tag{45}$$

which proves the inductive step for the sensitivity matrix on $\mathbf{V}$. We are now ready to prove the final part. Starting from equation 42, plugging in equation 43, we get

$$\left|\frac{\partial \mathbf{x}_v^{\ell+1,\alpha}}{\partial \mathbf{x}_u^{0,\beta}}\right| \leq \left(\left\|\frac{\partial \mathbf{x}_v^\ell}{\partial \mathbf{x}_u^0}\right\|_{L^1} + h\left|\frac{\partial \mathbf{v}_v^{\ell+1,\alpha}}{\partial \mathbf{x}_w^{\ell,\gamma}}\right|\left\|\frac{\partial \mathbf{x}_w^\ell}{\partial \mathbf{x}_u^0}\right\|_{L^1} + h\left|\frac{\partial \mathbf{v}_v^{\ell+1,\alpha}}{\partial \mathbf{v}_w^{\ell,\gamma}}\right|\left\|\frac{\partial \mathbf{v}_w^\ell}{\partial \mathbf{x}_u^0}\right\|_{L^1}\right)$$
$$\leq \left\|\frac{\partial \mathbf{x}_v^\ell}{\partial \mathbf{x}_u^0}\right\|_{L^1} + h(\delta_{vw}hw(N + 1) + wh\mathbf{A}_{vw})\left\|\frac{\partial \mathbf{x}_w^\ell}{\partial \mathbf{x}_u^0}\right\|_{L^1} + h(1 + hk)\delta_{vw}\left\|\frac{\partial \mathbf{v}_w^\ell}{\partial \mathbf{x}_u^0}\right\|_{L^1}$$
$$\leq (wd)^\ell\left(((1 + h + h^2(N + k + 1))\mathbf{I} + h^2\mathbf{A})^\ell\right)_{vu} +$$
$$+ h(\delta_{vw}hw(N + 1) + wh\mathbf{A}_{vw})(wd)^l\left(((1 + h + h^2(N + k + 1))\mathbf{I} + h^2\mathbf{A})^\ell\right)_{wu} +$$
$$+ \delta_{vw}wh(1 + hk)(wd)^\ell\left(((1 + h(N + k + 1))\mathbf{I} + h\mathbf{A})^\ell\right)_{wu}$$

$$\tag{46}$$

Summing over the last two terms and discarding the terms $O(h^3)$, we have

$$\left| \frac{\partial \mathbf{x}_v^{\ell+1,\alpha}}{\partial \mathbf{x}_u^{0,\beta}} \right| \leq (wd)^\ell \left( \left( (1+h+h^2(N+k+1))\mathbf{I} + h^2 \mathbf{A} \right)^\ell \right)_{vu} +$$
$$+ (wd)^\ell wh((1+h(N+k+1))\delta_{vw} + h\mathbf{A}_{vw}) \left( \left( (1+h+h^2(N+k+1))\mathbf{I} + h^2 \mathbf{A} \right)^\ell \right)_{wu} \tag{47}$$

Now we put $\delta_{vw}$ in front over the first term (and sum over $w$) to join the two terms together obtaining

$$\left| \frac{\partial \mathbf{x}_v^{\ell+1,\alpha}}{\partial \mathbf{x}_u^{0,\beta}} \right| \leq \delta_{vw}(wd)^\ell \left( \left( (1+h+h^2(N+k+1))\mathbf{I} + h^2 \mathbf{A} \right)^\ell \right)_{vw} +$$
$$+ (wd)^\ell wh((1+h(N+k+1))\delta_{vw} + h\mathbf{A}_{vw}) \left( \left( (1+h+h^2(N+k+1))\mathbf{I} + h^2 \mathbf{A} \right)^\ell \right)_{wu}$$
$$= w^{\ell+1}d^\ell((1+h+h^2(N+k+1))\delta_{vw} + h^2\mathbf{A}_{vw}) \left( \left( (1+h+h^2(N+k+1))\mathbf{I} + h^2 \mathbf{A} \right)^\ell \right)_{wu}$$
$$= w^{\ell+1}d^\ell \left( \left( (1+h^2(N+k+1))\mathbf{I} + h^2 \mathbf{A} \right)^{\ell+1} \right)_{vu} \tag{48}$$

Summing over $\beta$ gives the additional factor $d$, while maximizing over $\alpha$ gives us the thesis. $\qquad \square$

## C   Additional Results And Comparisons

### C.1   Extended Comparison

To further evaluate the performance of SONAR, we report a more complete comparison for the LRGB tasks in Table 7 and for the heterophilic tasks in Table 8. Specifically, in the LRGB setting, we include more multi-hop DGNs and ablate on the scores obtained with the original setting from [24] and the one proposed in [82]. The latter incorporates added residual connections and 3-layers MLP decoder. In the heterophilic setting, we include more MPNN-based models, graph transformers, and heterophily-designated GNNs. In both tables, we color the top three methods. Different from the main body of the paper, here we also include sub-variants of methods in the highlighted results, providing an additional perspective on the findings. Notably, our SONAR achieves state-of-the-art performance across all considered tasks.

### C.2   Ablations

To better understand the contribution of each component in SONAR, we conduct a series of ablation studies.

**Adaptive Resistance, Dissipation, and External Forces.**   We analyze the role of the dissipation mechanism, the external force term, and the adaptive edge resistance in shaping the model's dynamics and performance. Each of these components plays a crucial role in controlling information propagation over the graph. By systematically removing these elements, we assess their individual impact on long-range information flow and overall predictive performance.

Table 9 reports the mean and standard deviations on the test set for some tasks we analyzed in Section 4, i.e., graph transfer line-50, SSSP, Peptides-struct, and Roman-Empire. We report the results for each possible combination of adaptive resistance (i.e., adaptively re-computing the edge resistance at each step), dissipative, and external forcing components. We note that the importance of each component in SONAR varies depending on the task and the nature of the information it involves. In the graph transfer task, all components of SONAR are essential, particularly dissipation and external forcing as intermediate nodes contain random features that must be effectively filtered out. In the SSSP task, re-computing the resistances at each step leads to a performance drop, indicating that a constant propagation pattern is preferable. For `peptides-struct`, the external forcing term proves to be the most impactful, while dissipation offers little benefit. The best results on the `Roman-Empire` dataset are achieved using the purely conservative form of SONAR, highlighting the value of a non-dissipative signal propagation in that setting.

Lastly, to understand the importance of the energy preservation behavior of our SONAR, we compare its performance with that of a standard GCN, which can be considered as the most comparable baselines to our proposed SONAR. Both rely on the Laplacian operator and message-passing paradigm,

Table 7: Results for Peptides-func, Peptides-struct and PascalVOC-SP averaged over 3 training seeds. Baseline results are taken from [24, 46, 51, 82, 65, 21, 40, 52, 21]. Re-evaluated methods employ the 3-layer MLP readout proposed in [82]. Note that all MPNN-based methods include structural and positional encoding. The **first**, **second**, and **third** best scores are colored. ‡ means 3-layer MLP readout and residual connections are employed.

| Model | Peptides-func AP ↑ | Peptides-struct MAE ↓ | Pascal VOC-SP F1 ↑ |
|---|---|---|---|
| **MPNNs** | | | |
| GatedGCN | $58.64_{\pm 0.77}$ | $0.3420_{\pm 0.0013}$ | $0.2873_{\pm 0.0219}$ |
| GCN | $59.30_{\pm 0.23}$ | $0.3496_{\pm 0.0013}$ | $0.1268_{\pm 0.0060}$ |
| GCNII | $55.43_{\pm 0.78}$ | $0.3471_{\pm 0.0010}$ | $0.1698_{\pm 0.0080}$ |
| GINE | $54.98_{\pm 0.79}$ | $0.3547_{\pm 0.0045}$ | $0.1265_{\pm 0.0076}$ |
| **Multi-hop GNNs** | | | |
| DIGL+MPNN | $64.69_{\pm 0.19}$ | $0.3173_{\pm 0.0007}$ | $0.2824_{\pm 0.0039}$ |
| DIGL+MPNN+LapPE | $68.30_{\pm 0.26}$ | $0.2616_{\pm 0.0018}$ | $0.2921_{\pm 0.0038}$ |
| DRew-GCN | $69.96_{\pm 0.76}$ | $0.2781_{\pm 0.0028}$ | $0.1848_{\pm 0.0107}$ |
| DRew-GCN+LapPE | $\mathbf{71.50}_{\pm 0.44}$ | $0.2536_{\pm 0.0015}$ | $0.1851_{\pm 0.0092}$ |
| DRew-GIN | $69.40_{\pm 0.74}$ | $0.2799_{\pm 0.0016}$ | $0.2719_{\pm 0.0043}$ |
| DRew-GIN+LapPE | $\mathbf{71.26}_{\pm 0.45}$ | $0.2606_{\pm 0.0014}$ | $0.2692_{\pm 0.0059}$ |
| DRew-GatedGCN | $67.33_{\pm 0.94}$ | $0.2699_{\pm 0.0018}$ | $0.3214_{\pm 0.0021}$ |
| DRew-GatedGCN+LapPE | $69.77_{\pm 0.26}$ | $0.2539_{\pm 0.0007}$ | $0.3314_{\pm 0.0024}$ |
| GRED | $70.85_{\pm 0.27}$ | $0.2503_{\pm 0.0019}$ | – |
| GRED+LapPE | $\mathbf{71.33}_{\pm 0.11}$ | $\mathbf{0.2455}_{\pm 0.0013}$ | – |
| MixHop-GCN | $65.92_{\pm 0.36}$ | $0.2921_{\pm 0.0023}$ | $0.2506_{\pm 0.0133}$ |
| MixHop-GCN+LapPE | $68.43_{\pm 0.49}$ | $0.2614_{\pm 0.0023}$ | $0.2218_{\pm 0.0174}$ |
| **Transformers** | | | |
| GraphGPS+LapPE | $65.35_{\pm 0.41}$ | $0.2500_{\pm 0.0005}$ | $0.3748_{\pm 0.0109}$ |
| Graph ViT | $69.42_{\pm 0.75}$ | $\mathbf{0.2449}_{\pm 0.0016}$ | – |
| GRIT | $69.88_{\pm 0.82}$ | $\mathbf{0.2460}_{\pm 0.0012}$ | – |
| SAN+LapPE | $63.84_{\pm 1.21}$ | $0.2683_{\pm 0.0043}$ | $0.3230_{\pm 0.0039}$ |
| Transformer+LapPE | $63.26_{\pm 1.26}$ | $0.2529_{\pm 0.0016}$ | $0.2694_{\pm 0.0098}$ |
| **Modified and Re-evaluated**‡ | | | |
| GCN | $68.60_{\pm 0.50}$ | $\mathbf{0.2460}_{\pm 0.0007}$ | $0.2078_{\pm 0.0031}$ |
| GINE | $66.21_{\pm 0.67}$ | $0.2473_{\pm 0.0017}$ | $0.2718_{\pm 0.0054}$ |
| GatedGCN | $67.65_{\pm 0.47}$ | $0.2477_{\pm 0.0009}$ | $0.3880_{\pm 0.0040}$ |
| DRew-GCN+LapPE | $69.45_{\pm 0.21}$ | $0.2517_{\pm 0.0011}$ | – |
| GraphGPS+LapPE | $65.34_{\pm 0.91}$ | $0.2509_{\pm 0.0014}$ | $\mathbf{0.4440}_{\pm 0.0064}$ |
| **DE-GNNs** | | | |
| GRAND | $57.89_{\pm 0.62}$ | $0.3418_{\pm 0.0015}$ | $0.1918_{\pm 0.0097}$ |
| GraphCON | $60.22_{\pm 0.68}$ | $0.2778_{\pm 0.0018}$ | $0.2108_{\pm 0.0091}$ |
| A-DGN | $59.75_{\pm 0.44}$ | $0.2874_{\pm 0.0021}$ | $0.2349_{\pm 0.0054}$ |
| SWAN | $67.51_{\pm 0.39}$ | $0.2485_{\pm 0.0009}$ | $0.3192_{\pm 0.0250}$ |
| PH-DGN‡ | $70.12_{\pm 0.45}$ | $0.2465_{\pm 0.0020}$ | – |
| **Ours** | | | |
| SONAR | $68.42_{\pm 0.11}$ | $0.2525_{\pm 0.0038}$ | $\mathbf{0.4058}_{\pm 0.0039}$ |
| SONAR+LapPE | $70.47_{\pm 0.41}$ | $0.2486_{\pm 0.0006}$ | $\mathbf{0.4082}_{\pm 0.0037}$ |

Table 8: Mean test set score and std averaged over 4 random weight initializations on heterophilic datasets. The higher, the better. **First**, second, and **third** best results for each task are color-coded.

| Model | Roman-empire Acc ↑ | Amazon-ratings Acc ↑ | Minesweeper AUC ↑ | Tolokers AUC ↑ | Questions AUC ↑ |
|---|---|---|---|---|---|
| **[64]** | | | | | |
| MLP-2 | $66.04_{\pm0.71}$ | $49.55_{\pm0.81}$ | $50.92_{\pm1.25}$ | $74.58_{\pm0.75}$ | $69.97_{\pm1.16}$ |
| SGC-1 | $44.60_{\pm0.52}$ | $40.69_{\pm0.42}$ | $82.04_{\pm0.77}$ | $73.80_{\pm1.35}$ | $71.06_{\pm0.92}$ |
| MLP-1 | $64.12_{\pm0.61}$ | $38.60_{\pm0.41}$ | $50.59_{\pm0.83}$ | $71.89_{\pm0.82}$ | $70.33_{\pm0.96}$ |
| **Graph-agnostic** | | | | | |
| ResNet | $65.88_{\pm0.38}$ | $45.90_{\pm0.52}$ | $50.89_{\pm1.39}$ | $72.95_{\pm1.06}$ | $70.34_{\pm0.76}$ |
| ResNet+SGC | $73.90_{\pm0.51}$ | $50.66_{\pm0.48}$ | $70.88_{\pm0.90}$ | $80.70_{\pm0.97}$ | $75.81_{\pm0.96}$ |
| ResNet+adj | $52.25_{\pm0.40}$ | $51.83_{\pm0.57}$ | $50.42_{\pm0.83}$ | $78.78_{\pm1.11}$ | $75.77_{\pm1.24}$ |
| **MPNNs** | | | | | |
| GAT | $80.87_{\pm0.30}$ | $49.09_{\pm0.63}$ | $92.01_{\pm0.68}$ | $83.70_{\pm0.47}$ | $77.43_{\pm1.20}$ |
| GAT-sep | $\mathbf{88.75}_{\pm0.41}$ | $52.70_{\pm0.62}$ | $93.91_{\pm0.35}$ | $83.78_{\pm0.43}$ | $76.79_{\pm0.71}$ |
| GAT (LapPE) | $84.80_{\pm0.46}$ | $44.90_{\pm0.73}$ | $93.50_{\pm0.54}$ | $\mathbf{84.99}_{\pm0.54}$ | $76.55_{\pm0.84}$ |
| GAT (RWSE) | $86.62_{\pm0.53}$ | $48.58_{\pm0.41}$ | $92.53_{\pm0.65}$ | $85.02_{\pm0.67}$ | $77.83_{\pm1.22}$ |
| GAT (DEG) | $85.51_{\pm0.56}$ | $51.65_{\pm0.60}$ | $93.04_{\pm0.62}$ | $84.22_{\pm0.81}$ | $77.10_{\pm1.23}$ |
| Gated-GCN | $74.46_{\pm0.54}$ | $43.00_{\pm0.32}$ | $87.54_{\pm1.22}$ | $77.31_{\pm1.14}$ | $76.61_{\pm1.13}$ |
| GCN | $73.69_{\pm0.74}$ | $48.70_{\pm0.63}$ | $89.75_{\pm0.52}$ | $83.64_{\pm0.67}$ | $76.09_{\pm1.27}$ |
| GCN (LapPE) | $83.37_{\pm0.55}$ | $44.35_{\pm0.36}$ | $94.26_{\pm0.49}$ | $84.95_{\pm0.78}$ | $77.79_{\pm1.34}$ |
| GCN (RWSE) | $84.84_{\pm0.55}$ | $46.40_{\pm0.55}$ | $93.84_{\pm0.48}$ | $85.11_{\pm0.77}$ | $77.81_{\pm1.40}$ |
| GCN (DEG) | $84.21_{\pm0.47}$ | $50.01_{\pm0.69}$ | $\mathbf{94.14}_{\pm0.50}$ | $82.51_{\pm0.83}$ | $76.96_{\pm1.21}$ |
| SAGE | $85.74_{\pm0.67}$ | $53.63_{\pm0.39}$ | $93.51_{\pm0.57}$ | $82.43_{\pm0.44}$ | $76.44_{\pm0.62}$ |
| **Graph Transformers** | | | | | |
| Exphormer | $89.03_{\pm0.37}$ | $53.51_{\pm0.46}$ | $90.74_{\pm0.53}$ | $83.77_{\pm0.78}$ | $73.94_{\pm1.06}$ |
| NAGphormer | $74.34_{\pm0.77}$ | $51.26_{\pm0.72}$ | $84.19_{\pm0.66}$ | $78.32_{\pm0.95}$ | $68.17_{\pm1.53}$ |
| GOAT | $71.59_{\pm1.25}$ | $44.61_{\pm0.50}$ | $81.09_{\pm1.02}$ | $83.11_{\pm1.04}$ | $75.76_{\pm1.66}$ |
| GPS | $82.00_{\pm0.61}$ | $\mathbf{53.10}_{\pm0.42}$ | $90.63_{\pm0.67}$ | $83.71_{\pm0.48}$ | $71.73_{\pm1.47}$ |
| GPS$_{GCN+Performer}$ (LapPE) | $83.96_{\pm0.53}$ | $48.20_{\pm0.67}$ | $93.85_{\pm0.41}$ | $84.72_{\pm0.77}$ | $77.85_{\pm1.25}$ |
| GPS$_{GCN+Performer}$ (RWSE) | $84.72_{\pm0.65}$ | $48.08_{\pm0.85}$ | $92.88_{\pm0.50}$ | $84.81_{\pm0.86}$ | $76.45_{\pm1.51}$ |
| GPS$_{GCN+Performer}$ (DEG) | $83.38_{\pm0.68}$ | $48.93_{\pm0.47}$ | $93.60_{\pm0.47}$ | $80.49_{\pm0.97}$ | $74.24_{\pm1.18}$ |
| GPS$_{GAT+Performer}$ (LapPE) | $85.93_{\pm0.52}$ | $48.86_{\pm0.38}$ | $92.62_{\pm0.79}$ | $84.62_{\pm0.54}$ | $76.71_{\pm0.98}$ |
| GPS$_{GAT+Performer}$ (RWSE) | $87.04_{\pm0.58}$ | $49.92_{\pm0.68}$ | $91.08_{\pm0.58}$ | $84.38_{\pm0.91}$ | $77.14_{\pm1.49}$ |
| GPS$_{GAT+Performer}$ (DEG) | $85.54_{\pm0.58}$ | $51.03_{\pm0.60}$ | $91.52_{\pm0.46}$ | $82.45_{\pm0.89}$ | $76.51_{\pm1.19}$ |
| GPS$_{GCN+Transformer}$ (LapPE) | OOM | OOM | $91.82_{\pm0.41}$ | $83.51_{\pm0.93}$ | OOM |
| GPS$_{GCN+Transformer}$ (RWSE) | OOM | OOM | $91.17_{\pm0.51}$ | $83.53_{\pm1.06}$ | OOM |
| GPS$_{GCN+Transformer}$ (DEG) | OOM | OOM | $91.76_{\pm0.61}$ | $80.82_{\pm0.95}$ | OOM |
| GPS$_{GAT+Transformer}$ (LapPE) | OOM | OOM | $92.29_{\pm0.61}$ | $84.70_{\pm0.56}$ | OOM |
| GPS$_{GAT+Transformer}$ (RWSE) | OOM | OOM | $90.82_{\pm0.56}$ | $84.01_{\pm0.96}$ | OOM |
| GPS$_{GAT+Transformer}$ (DEG) | OOM | OOM | $91.58_{\pm0.56}$ | $81.89_{\pm0.85}$ | OOM |
| GT | $86.51_{\pm0.73}$ | $51.17_{\pm0.66}$ | $91.85_{\pm0.76}$ | $83.23_{\pm0.64}$ | $\mathbf{77.95}_{\pm0.68}$ |
| GT-sep | $87.32_{\pm0.39}$ | $52.18_{\pm0.80}$ | $92.29_{\pm0.47}$ | $82.52_{\pm0.92}$ | $78.05_{\pm0.93}$ |
| **Heterophily-Designated GNNs** | | | | | |
| CPGNN | $63.96_{\pm0.62}$ | $39.79_{\pm0.77}$ | $52.03_{\pm5.46}$ | $73.36_{\pm1.01}$ | $65.96_{\pm1.95}$ |
| FAGCN | $65.22_{\pm0.56}$ | $44.12_{\pm0.30}$ | $88.17_{\pm0.73}$ | $77.75_{\pm1.05}$ | $77.24_{\pm1.26}$ |
| FSGNN | $79.92_{\pm0.56}$ | $52.74_{\pm0.83}$ | $90.08_{\pm0.70}$ | $82.76_{\pm0.61}$ | $78.86_{\pm0.92}$ |
| GBK-GNN | $74.57_{\pm0.47}$ | $45.98_{\pm0.71}$ | $90.85_{\pm0.58}$ | $81.01_{\pm0.67}$ | $74.47_{\pm0.86}$ |
| GloGNN | $59.63_{\pm0.69}$ | $36.89_{\pm0.14}$ | $51.08_{\pm1.23}$ | $73.39_{\pm1.17}$ | $65.74_{\pm1.19}$ |
| GPR-GNN | $64.85_{\pm0.27}$ | $44.88_{\pm0.34}$ | $86.24_{\pm0.61}$ | $72.94_{\pm0.97}$ | $55.48_{\pm0.91}$ |
| H2GCN | $60.11_{\pm0.52}$ | $36.47_{\pm0.23}$ | $89.71_{\pm0.31}$ | $73.35_{\pm1.01}$ | $63.59_{\pm1.46}$ |
| JacobiConv | $71.14_{\pm0.42}$ | $43.55_{\pm0.48}$ | $89.66_{\pm0.40}$ | $68.66_{\pm0.65}$ | $73.88_{\pm1.16}$ |
| **Our** | | | | | |
| SONAR | $89.82_{\pm0.57}$ | $52.22_{\pm0.14}$ | $96.29_{\pm0.73}$ | $83.57_{\pm1.44}$ | $74.96_{\pm1.10}$ |

Table 9: Mean test set results with standard deviations for different combinations of SONAR components. The last line presents the results from a standard GCN for the comparison with a non-conservative method. **First**, **second**, and **third** best results for each task are color-coded.

| Adaptive Resistace | Dissipation | External Forcing | Line-50 $\log_{10}(\text{MSE}) \downarrow$ | SSSP $\log_{10}\text{MSE} \downarrow$ | Peptides-struct MAE $\downarrow$ | Roman-Empire Acc $\uparrow$ |
|---|---|---|---|---|---|---|
| ✓ | ✓ | ✓ | **-6.4069**$_{\pm 0.4653}$ | -3.0213$_{\pm 0.2431}$ | 0.2611$_{\pm 0.0075}$ | 88.24$_{\pm 0.24}$ |
| ✓ | ✓ | − | -5.7542$_{\pm 0.6264}$ | -3.8182$_{\pm 0.2239}$ | 0.2592$_{\pm 0.0168}$ | 88.61$_{\pm 1.11}$ |
| ✓ | − | ✓ | **-6.2545**$_{\pm 0.3354}$ | -2.7857$_{\pm 0.1731}$ | **0.2506**$_{\pm 0.0010}$ | **89.27**$_{\pm 0.77}$ |
| ✓ | − | − | -5.3827$_{\pm 0.4078}$ | -2.8108$_{\pm 0.2010}$ | 0.2560$_{\pm 0.0098}$ | **89.81**$_{\pm 0.57}$ |
| − | ✓ | ✓ | **-6.3598**$_{\pm 0.5403}$ | **-6.7515**$_{\pm 0.0589}$ | 0.2631$_{\pm 0.0052}$ | 88.95$_{\pm 0.57}$ |
| − | ✓ | − | -5.9058$_{\pm 0.5576}$ | **-6.5939**$_{\pm 0.2925}$ | 0.2508$_{\pm 0.0036}$ | 89.12$_{\pm 0.82}$ |
| − | − | ✓ | -4.9749$_{\pm 0.2777}$ | **-6.5241**$_{\pm 0.5738}$ | **0.2486**$_{\pm 0.0006}$ | 89.22$_{\pm 0.41}$ |
| − | − | − | -5.4487$_{\pm 0.7370}$ | -6.3226$_{\pm 0.1126}$ | **0.2503**$_{\pm 0.0006}$ | **89.42**$_{\pm 0.67}$ |
| GCN results for comparison | | | -3.5032$_{\pm 0.0001}$ | 0.9499$_{\pm 0.0001}$ | 0.3496$_{\pm 0.0013}$ | 73.69$_{\pm 0.74}$ |

Table 10: Mean train and test errors of identical SONAR models with different step sizes $h$ on the Ring-10 task from Section 4.1.

| step size $h$ | mean train loss | mean test loss |
|---|---|---|
| 0.05 | 0.0020273 | 0.0019732 |
| 0.10 | 0.0019947 | 0.0019512 |
| 0.50 | 0.0039039 | 0.0039458 |
| 1.00 | 0.0085834 | 0.0086960 |

but GCN lacks energy preservation guarantees for long-range information propagation, adaptive resistance, and external forces. We note that GCN fails in solving these tasks since it performs a dissipative signal propagation. As a result, information about distant nodes becomes increasingly indistinguishable and ultimately lost. In contrast, our SONAR successfully solves these tasks because it performs a non-dissipative propagation. This is enabled by explicitly preserving the energy of the system (Equation (9)) in its design, with the result of preserving signals during propagation (as theoretically discussed in Section 3).

Overall, these results highlight SONAR's ability to balance conservative and non-conservative behaviors, enhancing its ability to learn more complex and task-specific dynamics. We believe that the modular design of SONAR, with components that can be selectively activated, provides high flexibility and allows SONAR to adapt to a wide range of scenarios while maintaining computational efficiency, without relying on global attention mechanisms or graph rewiring.

This analysis also suggests a practical guideline: for propagation-heavy tasks, forcing (and often dissipation) is beneficial; dissipation is particularly useful under high noise (e.g., graph transfer tasks); and adaptive resistance tends to help particularly with heterophilic tasks. These trends narrow the hyperparameter search space, though a small hyperparameter search is still necessary for optimal performance.

**The Effect of the Step-Size $h$.** Since the step size controls the numerical accuracy of the wave equation, its careful selection is essential for preserving SONAR's properties and performance. As with classical numerical solvers for physical equations, a high number of iterations requires a smaller step size, and vice versa. In practice, we observed that small variations in the step size have little impact on performance, and exploring different orders of magnitude is sufficient to identify near-optimal values. For a practical example, we consider the ring-10 task described in Section 4.1 and train an identical configuration of SONAR with 4 different step sizes. The results, reported in Table 10, indicate that close step sizes produce comparable performance, with noticeable differences arising only for large changes. This demonstrates that SONAR's behavior is robust to small changes in the step size.

**Runtimes.** Additionally, we report runtime analyses to evaluate the computational efficiency of SONAR with respect to state-of-the-art methods in Table 11. Specifically, we report the training and inference times in milliseconds, as well as the memory usage obtained on the Roman-Empire dataset, fixing the model trainable parameters to 100k. As can be seen from the results in the Table, our

SONAR maintains a similar runtime and memory consumption to GCN, which has linear complexity with respect to the graph size. All runtimes are measured on an NVIDIA H100 GPU.

Table 11: Training and inference time (in milliseconds) and memory usage on the Roman-Empire dataset measured on a NVIDIA H100 GPU. All models have approximately 100k trainable parameters; GPS uses 2 attention heads.

| | | Depth | | | |
|---|---|---|---|---|---|
| **Method** | **Metrics** | **4** | **8** | **16** | **32** |
| GCN | Training (ms) | $3.32_{\pm 0.07}$ | $5.08_{\pm 0.85}$ | $7.99_{\pm 0.3}$ | $13.04_{\pm 0.12}$ |
| | Inference (ms) | $1.98_{\pm 0.02}$ | $3.24_{\pm 0.06}$ | $5.0_{\pm 0.07}$ | $8.05_{\pm 0.04}$ |
| | Training Mem (GB) | 0.25 | 0.25 | 0.28 | 0.34 |
| | Inference Mem (GB) | 0.21 | 0.18 | 0.16 | 0.14 |
| GPS | Training (ms) | $101.21_{\pm 0.25}$ | $321.33_{\pm 1.27}$ | $642.53_{\pm 0.54}$ | OOM |
| | Inference (ms) | $88.51_{\pm 0.14}$ | $157.31_{\pm 0.22}$ | $287.27_{\pm 0.15}$ | OOM |
| | Training Mem (GB) | 0.35 | 43.59 | 75.13 | OOM |
| | Inference Mem (GB) | 15.79 | 15.78 | 15.78 | OOM |
| SONAR (1 block) | Training (ms) | $10.11_{\pm 0.97}$ | $18.01_{\pm 1.11}$ | $32.62_{\pm 0.12}$ | $63.18_{\pm 0.82}$ |
| | Inference (ms) | $3.78_{\pm 0.04}$ | $6.59_{\pm 0.02}$ | $12.23_{\pm 0.03}$ | $23.45_{\pm 0.09}$ |
| | Training Mem (GB) | 0.71 | 1.24 | 2.29 | 4.41 |
| | Inference Mem (GB) | 0.27 | 0.28 | 0.28 | 0.28 |
| SONAR (2 blocks) | Training (ms) | $9.63_{\pm 0.18}$ | $16.21_{\pm 0.1}$ | $29.62_{\pm 0.09}$ | $56.56_{\pm 0.47}$ |
| | Inference (ms) | $4.03_{\pm 1.14}$ | $5.87_{\pm 0.05}$ | $10.47_{\pm 0.02}$ | $19.74_{\pm 0.07}$ |
| | Training Mem (GB) | 0.57 | 0.97 | 1.77 | 3.37 |
| | Inference Mem (GB) | 0.23 | 0.23 | 0.23 | 0.23 |

# D   Limitations

This paper contributes to the field of machine learning by advancing (differential equation-inspired) graph neural networks, with a focus on enhancing long-range information propagation. Specifically, we introduce a novel methodology that enables provable long-range propagation on graphs, grounded in a physically interpretable model based on wave dynamics and their underlying properties.

While we believe our work offers a meaningful contribution, it is important to clarify that SONAR is designed to address the long-range propagation problem in GNNs through the lens of DE-GNNs. Therefore, our model prioritizes on problems that require effective modeling of long-range interaction while maintaining low computational overhead. However, if the goal is to maximize downstream performance regardless of computational complexity, alternative approaches such as multi-hop GNNs or graph transformers may also be appropriate.

A second limitation arises from SONAR's formulation as a discrete approximation of a continuous dynamical system. Its behavior depends on the discretization scheme, making the choice of the step size $h$ crucial. As shown in Theorem 3.4, the sensitivity bound can grow exponentially with the number of steps, potentially leading to exploding gradients if $h$ is not appropriately tuned. We emphasize, however, that this issue can be effectively mitigated through careful step size selection and was never encountered in our experiments.

