# OpenReview forum: "SONAR: Long-Range Graph Propagation Through Information Waves"
_NeurIPS.cc/2025/Conference — NeurIPS 2025 poster_

### Official Review · Reviewer_byZL · 2025-07-02

**Clarity:** 3
**Significance:** 3
**Originality:** 3
**Rating:** 4
**Confidence:** 4

**Summary:**

This paper introduces SONAR, a GNN architecture designed to address the challenge of long-range information propagation. SONAR models information flow on graphs using the wave equation, which allows signals to travel over long distances as oscillations with minimal energy loss. The authors provide the theoretical backing, proving energy conservation and non-vanishing sensitivity between distant nodes. Extensive experiments on synthetic, long-range, and heterophilic benchmarks demonstrate that SONAR achieves state-of-the-art performance, particularly on tasks that require modeling distant interactions.

**Questions:**

1. The theoretical guarantees rely on specific discretization schemes and parameters like step size $h$. How sensitive is the model's performance to these choices?

**Ethical Concerns:**

["NO or VERY MINOR ethics concerns only"]

**Final Justification:**

Comprehensively considering the paper's contribution and the rebuttal, I think my original score is reasonable.

**Limitations:**

yes

**Quality:**

3

**Strengths And Weaknesses:**

- Strengths
1. Grounding the GNN in the wave equation provides a principled foundation for long-range propagation that is both physically intuitive and mathematically sound.
2. SONAR shows dominant performance on synthetic long-range tasks and is highly competitive on challenging real-world benchmarks.

- Weaknesses
1. SONAR relies heavily on discretization schemes and parameter tuning, such as step size $h$, which may influence stability and sensitivity bounds.

---

> ### Author Rebuttal · Authors · 2025-07-30
>
> We thank the reviewer for highlighting the quality of our model and its soundness both in physical and mathematical terms. We now address the comments in the review.
>
> We agree with the Reviewer that the kind of discretization and the integration scheme used to solve the wave equation numerically are important in the effectiveness of SONAR. Nonetheless, in our work, we use a simple Euler scheme, which straightforwardly showed very good performance with low effort. We believe that more complex schemes might increase the performance of SONAR even more, but we left those for future work.
>
> However, we argue that our theoretical results on the long-range capabilities and sensitivity remain true with other discretization schemes, with some minor corrections. Most importantly, the energy conservation Theorem 3.1 does not depend on a specific discretization and remains true as long as the node features evolve following our defined dynamics. The sensitivity results in Theorems 3.3 and 3.4 are cast on our specific choice for discretization, but could be easily adapted to other explicit schemes, following our proofs.
>
> In any case, the step size acts as a hyperparameter to be tuned, and our experimental results show that it is easy to control.
>
> About the sensitivity to $h$. Since the step size is a hyperparameter of our model and the quality of the solution to the wave equation depends on it, choosing a good step size is important to guarantee the properties and performance of our model. In general, a high number of iterations requires a lower step size, and vice versa, like classical numerical solvers for physical equations. In our practical tests, we saw that the performance does not change significantly with small variations of the step size, while it is enough to try different orders of magnitude to have optimal results. For a practical example, we considered “ring 10” within the graph transfer task. We trained an identical configuration of SONAR with 4 different step sizes, and we report the results in the table below.
> | step size $h$ | mean train loss | mean test loss |
> |---|:---:|:---:|
> | 0.05 | 0.0020273 | 0.0019732 |
> | 0.10 | 0.0019947 | 0.0019512 |
> | 0.50 | 0.0039039 | 0.00394583 |
> | 1.00 | 0.0085834 | 0.0086960 |
>
> As we can see, similar values produce similar results, without drops in performance, with differences appearing only with large changes in the magnitude of $h$.

---

> > ### Comment · Reviewer_byZL · 2025-08-05
> > **Response to Authors**
> >
> > Thanks for your detailed responses which have addressed most my concerns. Comprehensively considering the paper and the rebuttal, I think my original score is reasonable. I vote for accepting this paper.

---

### Official Review · Reviewer_jhtz · 2025-07-03

**Clarity:** 3
**Significance:** 2
**Originality:** 2
**Rating:** 4
**Confidence:** 4

**Summary:**

SONAR is a differential-equation GNN that treats message passing as a wave equation on graphs. Each edge carries an adaptive resistance, and state-dependent damping/forcing terms let the network interpolate between fully conservative (energy-preserving) and dissipative dynamics. The authors prove that, in the conservative limit, SONAR maintains a constant total feature energy and its node-to-node sensitivity doesn't vanish, thereby guaranteeing non-degenerate long-range propagation. A finite-difference discretisation is wrapped in learnable MLP blocks. Empirically, SONAR achieves strong results across synthetic datasets, long-range tasks, and heterophilic datasets.

**Questions:**

Please see Weaknesses

**Ethical Concerns:**

["NO or VERY MINOR ethics concerns only"]

**Final Justification:**

In general, my concerns have been addressed by the rebuttal. Given the new additions during discussion, I would recommend acceptance.

**Limitations:**

yes

**Quality:**

2

**Strengths And Weaknesses:**

Pros:
- The wave-equation view and accompanying energy/sensitivity theorems provide good theoretical guarantees for long-range information flow in GNNs
- The model achieves strong empirical results across various benchmarks
- The complexity is sub-quadratic and more manageable compared with graph transformers

Cons:
- According to ablation studies, the introduced Adaptive Resistance and Dissipative/External Forcing terms do not seem to have a consistent effect on different datasets, making them almost pure grid search.
- While this paper derives bounds on the sensitivity between nodes and argues it is "well-controlled", to support the long-range modeling capacity, it would be better to analyze how the sensitivity changes with the distance between nodes (or the relative ratio between distant node's influence and close node's influence). This would reflect whether distant nodes are downplayed compared to closer nodes.
- This paper lacks a comparison/discussion of a relevant work GRED [1]. GRED also interleaves linear recurrence with MLPs, and seems to perform better on LRGB
    - [1] Recurrent Distance Filtering for Graph Representation Learning. ICML 2024.
- How would the recurrence length $L$ affect the performance?

---

> ### Author Rebuttal · Authors · 2025-07-30
>
> We thank the reviewer for the insightful review and for highlighting that our SONAR has good theoretical guarantees for long-range propagation in GNNs and that it achieves strong empirical results with manageable complexity. Below, we provide detailed responses to each of the comments. We hope that these clarifications and revisions are satisfactory, and that the Reviewer will consider revising their score.
>
> - Regarding the effects of additional components. The fact that the influence of the introduced additional components varies across tasks is indeed expected, as the specific nature of each task determines how information must be propagated through the graph to achieve optimal performance. For example, the graph transfer task requires propagating information between distant nodes while filtering out irrelevant information from other nodes. This demands strong long-range capabilities and the ability to selectively retain or discard information; therefore, all of SONAR’s additional components become essential. By contrast, in the SSSP task, adaptive resistance is not needed, as the goal is to compute distances where each edge contributes equally, effectively requiring uniform propagation across the graph. These results highlight SONAR’s ability to balance conservative and non-conservative behaviors, enhancing its ability to learn more complex and task-specific dynamics. We believe that the modular design of SONAR, with components that can be selectively activated, provides high flexibility and allows SONAR to adapt to a wide range of scenarios while maintaining computational efficiency, without relying on global attention mechanisms or graph rewiring.
> - Regarding node sensitivity:
> Theorems 3.3 and 3.4 show that, compared to a standard MPNN-like architecture as studied in [2], our SONAR has improved capabilities of long-range propagation between nodes. We note that these results are not specific to a particular distance and are true for any two nodes with a viable path in the graph.  To the best of our knowledge, no published work explicitly studies the sensitivity between two distant nodes as suggested by the Reviewer, and in a way different from [2,3], as this sensitivity is also inherently dependent on the specific topology of the graph.
>    Nonetheless, to accommodate the Reviewer’s comment, below we provide an empirical validation of our findings along the lines of the requested empirical sensitivity analysis. We considered “line 50” in the graph transfer task, where the topology and the task require the model to perform effective long-range communication, and where the conditions of the experiment can be fully controlled. We computed the norm of the sensitivity matrix between node 0 and nodes at different distances in the graph (10, 20, 30, 40, 50). We compare the results of SONAR ( with step size h=0.1) with those of a standard GCN
> | distance  | **SONAR sensitivity** | **GCN sensitivity** |
> |---|:---:|:---:|
> | 10 | 0.0413 | 1.15 $\times 10^{-5}$ |
> | 20 | 0.0334 | 1.04 $\times 10^{-5}$ |
> | 30 | 0.0616 | 0.526 $\times 10^{-5}$ |
> | 40 | 0.0253 | 0.0 |
> | 50 | 0.0324 | 0.0 |
>
>    As we can see, the sensitivity matrix norm for SONAR is stable across distances, thus it propagates more information consistently, confirming our theoretical results. GCN, on the other hand, struggles to propagate information effectively, as the sensitivity norm is 2 orders of magnitude lower than SONAR, and it becomes null at larger distances, indicating that distant nodes fail to communicate.
>
> - Regarding GRED. We thank the Reviewer for the reference that we included in our revised paper. We note that while the results from GRED [1] are marginally higher, they are still comparable to the ones from our SONAR. Nonetheless, we would like to recall that GRED is a higher-order method, as, at each iteration, each node is updated with the information coming from different hops of distance based on shortest path computation, which are then aggregated using additional models such as Linear Recurrent Networks. On the other hand, SONAR aggregates from a single-hop neighborhood, retaining MPNN-like complexity and achieving comparable performance.
>
> - Regarding L. The influence of the recurrence length L on performance is task-dependent, and in our experiments, L is treated as a hyperparameter. The underlying intuition is that L should match the distance between the nodes that need to communicate to solve the task; otherwise, the model will suffer from underreaching. In tasks like graph transfer, where the communication distance is known, L can be set directly without the need for extensive tuning. In contrast, when this distance is not known in advance, tuning L is necessary to optimize performance. Figure 2 clearly shows that the performance of SONAR does not degrade by increasing the distance between such nodes, and instead, when L is big, SONAR is the best performing.
>
> ---------------------------------
> In light of these clarifications and the additional experiments, we hope this addresses the Reviewer’s concerns and they can consider revising their score.
>
>
> [1] Recurrent Distance Filtering for Graph Representation Learning. ICML 2024.
>
> [2] Topping et al. Understanding over-squashing and bottlenecks on graphs via curvature. In ICLR 2023
>
> [3] Di Giovanni et al. On over-squashing in message passing neural networks: the impact of width, depth, and topology. In ICML 2023

---

> > ### Author Response · Authors · 2025-08-05
> >
> > Dear Reviewer jhtz,
> >
> > Thank you once again for your detailed and constructive review of our paper.
> >
> > Since the rebuttal, we have made several additions based on your feedback. In particular:
> >
> > - We expanded our discussion on the roles of SONAR’s components, emphasizing how they enable both dissipative and conservative behaviors, thus supporting strong performance across diverse tasks.
> >
> > - We provided further discussion regarding node sensitivity as well as added a quantitative analysis of SONAR’s node sensitivity, reporting the norms across different node distances. This analysis confirms that SONAR exhibits significantly stronger long-range influence compared to standard GCN. Further details are provided in our discussion with Reviewer gMHH.
> >
> > - We clarified the differences with GRED, which is a higher-order model, showing that SONAR performs competitively while retaining MPNN-like complexity.
> >
> > - We discussed the impact of the hyperparameter $L$.
> >
> > We believe the inclusion of this discussion in the revised manuscript helps reinforce the strength of our contributions. We hope these clarifications and additions address your concerns and can lead to an increase in the score. Nevertheless, we remain available for any further questions.
> >
> > Best regards,
> >
> > The Authors

---

> > > ### Author Response · Authors · 2025-08-08
> > >
> > > Dear Reviewer jhtz,
> > >
> > > Given that we are less than 24 hours from the end of the discussion, we would like to thank the reviewer again for their thoughtful review. We have put significant effort into our rebuttal, carefully addressing all concerns and questions raised by the reviewers.
> > >
> > > We believe that our rebuttal has further strengthened our work, and we will incorporate these clarifications into the revised manuscript. We hope our rebuttal has contributed to raising your score toward acceptance of our work.
> > >
> > >
> > > Best Regards,
> > >
> > > The Authors

---

> > > > ### Comment · Reviewer_jhtz · 2025-08-09
> > > > **Thanks for the rebuttal**
> > > >
> > > > I appreciate the detailed response by the authors:
> > > >
> > > > - I understand the authors' claim that the proposed modular framework allows for flexible design, but is there a more principled way to determine what modules should be used for a new dataset (e.g., based on some statistics or simple metrics that can be obtained quickly)? Otherwise, we may still have to try every combination before we know what modules work
> > > > - Thanks for the additional results. They look great and are a very good addition to the paper. I suggest the authors include them in the updated version.
> > > > - Thanks for the clarification. I suggest the authors explicitly mention the relation/difference to GRED in related work and add its performance to the table
> > > > - OK. I understand some hyperparameter tuning is unavoidable.

---

> > > > > ### Author Response · Authors · 2025-08-09
> > > > >
> > > > > We thank the reviewer for engaging with us. We believe that these experiments and clarifications have further strengthened our work. Thus, following also the reviewer's suggestion, we will incorporate this discussion into the revised manuscript.
> > > > >
> > > > > We agree that automatic selection of modules would be ideal. However, as with standard deep-learning building blocks, real-world datasets involve many interacting factors (e.g., graph structure, noise, and task objective), which makes universal recommendations difficult. Our experiments suggest a pragmatic guideline: for propagation-heavy tasks, include forcing (and often dissipation); dissipation is more useful with a high level of noise (e.g., graph transfer tasks); and adaptive resistance tends to help particularly with heterophilic tasks. These trends substantially reduce the search space, though a small hyperparameter search is still necessary for optimal performance. We will include this insights in the revised manuscript. Investigating principled selection rules based on measurable graph statistics is an interesting direction for future work.
> > > > >
> > > > > We would like to thank the reviewer again for their thoughtful review. We are pleased that the reviewer appreciated our additional evaluations and clarifications. In light of this, we hope that the reviewer will consider raising the score in their final decision to reflect their positive view of the current version of our work.

---

### Official Review · Reviewer_gMHH · 2025-07-19

**Clarity:** 3
**Significance:** 3
**Originality:** 3
**Rating:** 4
**Confidence:** 4

**Summary:**

This paper introduces SONAR, a novel GNN architecture that models information flow using the graph wave equation. The goal is to address the over-squashing problem and enable effective long-range information propagation. The model incorporates adaptive resistances and external forces to balance conservative, energy-preserving dynamics with more flexible, non-conservative behaviors. The authors provide comprehensive theoretical analysis and well-grounded empirical results on several benchmarks to support their claims.

**Questions:**

- For disconnected nodes, is the adaptive resistance term a_uv always kept at 0, or can the model learn a non-zero weight, effectively creating a new edge?

- There appears to be a typo in Equation (6). Shouldn't the second line be  V˙(t)=−L_aX(t)W−D(X(t))⊙V(t)+F(X(t)) to align with Equation (5) and the discretized version in Equation (7)?

**Ethical Concerns:**

["NO or VERY MINOR ethics concerns only"]

**Final Justification:**

The author has successfully address the concerns I have for the rebuttal, mostly regarding the behaviour of SONAR over synthetic long-range task. I find the sensitivity analysis very useful. I would recommend acceptance of this paper providing that there are strong theoretical and empirical evidence of this paper.

**Limitations:**

yes

**Quality:**

3

**Strengths And Weaknesses:**

**Strengths**

- The motivation for using wave-based dynamics is very clear and well-articulated.
- The paper includes a complete theoretical derivation regarding over-squashing and information propagation, which is commendable.
- The writing is exceptionally clear and the paper is well-structured.
- The claims are supported by multiple experiments across a range of tasks.

**Weaknesses**

- **Lack of Qualitative Insight**: While the motivation is strong, the paper lacks insights into why the wave-like message passing works, and how is different than the exisitng GNN designed for long-range. The results section is simply a description of performance metrics rather than an analysis of the model's internal dynamics. I would appreciate if author could include some examples or case study to show how the proposed wave-like messages help improve the long-range, for example, in some synthetic datasets. This would be my major concern.

 - **Outdated/Questionable Benchmarks**: LRGB datasets, particularly Peptides-func and Peptides-struct, have been shown by recent work [1] to not be truly long-range tasks. The authors should consider including more convincing real-world experiments that truely require long-range, such as other datasets in LRGB. For heterophilic dataset, the comparison is missing more recent and relevant baselines designed for heterophily, such as CoGNN [2] or adaptive message passing neural network [3]. I would appreciate if the author could add additional experiments over these to support the claim that SONAR are working as expected in real-world experiments, but due to the benchmarking issues I still have some doubts over the performances of SONAR.

[1] Bamberger, Jacob, et al. "On Measuring Long-Range Interactions in Graph Neural Networks." ICML (2025).

[2] Finkelshtein, Ben, et al. "Cooperative graph neural networks." ICML (2024).

[3] Errica, Federico, et al. "Adaptive message passing: A general framework to mitigate oversmoothing, oversquashing, and underreaching." ICML (2025).

---

> ### Author Rebuttal · Authors · 2025-07-30
>
> We thank the Reviewer for the useful feedback and for recognizing the soundness of our work and our efforts to provide a theoretically grounded methodology while addressing information propagation and over-squashing problems in GNNs. We also appreciate their comments on the exceptional quality of our writing.
>
> We hope our clarifications and revisions will address their concerns and prompt a reconsideration of their evaluation and score.
>
> 1) Regarding more insights. We thank the Reviewer for the thoughtful comment. Compared to other GNNs designed for long-range propagation, SONAR achieves a non-dissipative behavior by design without requiring any weight regularization (as for ADGN or SWAN). Furthermore, SONAR propagates information long-range without rewiring techniques (as for DRew) or attention mechanisms (as for GAT or GPS), which are inefficient and computationally intensive. We would like to highlight that GCN-like models are the most comparable baselines to our proposed SONAR, as both leverage the graph Laplacian in their message-passing mechanisms. However, GCNs use the Laplacian directly, thus leading to the dissipative dynamics described by the equation $\dot{X} = LXW$. This formulation lacks guarantees for long-range information propagation. In contrast, SONAR is grounded in the wave equation on graphs [4]; thus, it follows the explicit non-dissipative dynamics defined by $\ddot{X} = -L_aXW$. This formulation ensures energy conservation and long-range propagation, as formally established by our theorems. Moreover, by integrating adaptive edge resistances and external forces, our method balances conservative and non-conservative behaviors, improving the ability to learn more complex dynamics. Regarding the inclusion of case studies, we would like to note that the synthetic tasks in Sections 4.1 and 4.2 already serve this purpose. Specifically, the tasks in Sec 4.1 are based on shortest path computations; thus, GNNs must heavily rely on global information and effectively learn to transport information across the entire graph. Tasks in Section 4.2 challenge models to transfer information from a source to a target node located at an increased distance. As shown in our results, SONAR consistently outperforms state-of-the-art models. To provide further insights into the long-range capabilities of SONAR in these controlled and synthetic tasks, we analysed in depth the long-range effects on the “line 50” graph in the transfer task. Here, the topology and the task require the model to perform effective long-range communication up to 50 hops. To verify long-term propagation of information, we studied the norm of the sensitivity matrix (Theorem 3.4) between node 0 and nodes at different distances in the graph (10, 20, 30, 40, 50). We compare the results of SONAR ( with step size h=0.1) with those of a standard GCN.
> | distance  | **SONAR sensitivity** | **GCN sensitivity** |
> |---|:---:|:---:|
> | 10 | 0.0413 | 1.15 $\times 10^{-5}$ |
> | 20 | 0.0334 | 1.04 $\times 10^{-5}$ |
> | 30 | 0.0616 | 0.526 $\times 10^{-5}$ |
> | 40 | 0.0253 | 0.0 |
> | 50 | 0.0324 | 0.0 |
>
>    As we can see, the sensitivity matrix norm for SONAR is stable across distances, thus it propagates more information consistently. GCN, on the other hand, struggles to propagate information effectively, as the sensitivity norm is 2 orders of magnitude lower than SONAR, and it becomes null at larger distances, indicating that distant nodes fail to communicate.
>    We believe this supports that the wave-like propagation implemented by SONAR helps in improving long-range propagation and that it provides deeper empirical insights and discussions, along the lines of what was highlighted by the reviewer. We included this discussion in the revised paper.
>
> 2) Regarding benchmarks:
> We thank the reviewer for the references. For our submission, we focused on the most relevant and widely used benchmarks for long-range propagation that were available at the time. Specifically, the work in  [1] became publicly available after the submission deadline. Nevertheless, to accommodate the Reviewer’s comment, we are now including results on the Pascal-VOC superpixel task, which, according to [1], is considered a long-range benchmark.
> | Model | PascalVOC (F1 score $\uparrow$) |
> |---|:---:|
> | **GPS** + LapPE | $0.451_{\pm 0.01}$ |
> | **GT** + LapPE | $0.392_{\pm 0.00}$ |
> | **Gated GCN** | $0.378_{\pm 0.00}$ |
> | **GINE** | $0.275_{\pm 0.00}$ |
> | **GCN+VN** + RWSE | $0.245_{\pm 0.00}$ |
> | **GCN** + RWSE | $0.185_{\pm 0.00}$ |
> | **DREW** (Gated GCN) + LapPE | $0.331_{\pm 0.00}$ |
> | **SONAR** | $0.376_{\pm 0.01}$ |
>
>
>    We remark that the results obtained by SONAR are without Positional Encoding.
>
>    Due to the limited time and the difficulty of the task, we were not able to perform a complete grid search for the optimal parameters of our model now (but this is running and will be included in the final version of the paper). However, our results are in line with the most expressive and long-range models, falling very close to Transformer models. We argue that this further shows the long-range capabilities of SONAR.
>
>    We would also like to emphasize that our paper already includes results on at least six other tasks requiring strong long-range capabilities, as discussed in our previous response. For example, the graph transfer tasks require propagating information up to 50 hops, providing further evidence of SONAR’s ability to propagate information over long distances. Finally, following up on the Reviewer’s comment, we have also included results from [2] and [3] in the revised version of the paper.
>
> [1] Bamberger, Jacob, et al. "On Measuring Long-Range Interactions in Graph Neural Networks." ICML (2025).
>
> [2] Finkelshtein, Ben, et al. "Cooperative graph neural networks." ICML (2024).
>
> [3] Errica, Federico, et al. "Adaptive message passing: A general framework to mitigate oversmoothing, oversquashing, and underreaching." ICML (2025).
>
> [4] Joel Friedman and Jean-Pierre Tillich. Wave equations for graphs and the edge-based Laplacian. Pacific J. Math., 216(2):229–266, October 2004
>
> About the questions:
> - Yes, the Reviewer is correct when saying that disconnected nodes always have a zero adaptive resistance. In particular, the resistance a_uv is calculated only between connected nodes. We clarified this aspect in the revised paper.
> - We thank the reviewer for noticing the typo; we fixed it in the revised paper.
>
> ---------------------------------
> In light of these clarifications and the additional experiments, we hope this addresses the Reviewer’s concerns and they can consider revising their score.

---

> > ### Comment · Reviewer_gMHH · 2025-08-03
> >
> > Thank author for the detailed rebuttal and additional experimentation. I have some follow up questions:
> > 1. How is the energy described in the paper related to dirichlet energy? The connection between dirichlet energy and over-smoothing has been drawn in [1].
> > 2. Even though authors claimed that section 4.1 and 4.2 already show the purpose of "inclusion of case study", I still think a detailed breakdown over, e.g., how SONAR benefits from energy perservation on one of the example in one of the task shown, is better than just listing number of improvements to help readers understand the benifit of the method. Can author detailed in one example, such as shortest path task or alike. how the learnt SONAR method can help solve the problem whereas the traditional GCN cannot?
> > 3. I appreciate author's additional experiments of the empirical sensitivity analysis and would suggest the author to add this into the main body of the paper. However, I notice that the sensitivity is not monotonically decreasing as the distances grow. Does this have practical impact? E.g., why would some tasks need these variation over different hops of neighbor? What are the rationale of in learning distance 50 is more senstive than distance 40?
> >
> >
> > I would be happy to raise my score if the author can convincingly address my follow up questions.
> >
> > [1] Di Giovanni, Francesco, et al. "Understanding convolution on graphs via energies." arXiv preprint arXiv:2206.10991 (2022).

---

> ### Author Response · Authors · 2025-08-04
> **Response to further questions (1/2)**
>
> We thank the Reviewer for the reply, and we are happy to answer the follow-up questions below.
>
>
> 1) The energy of the system presented in Eq. 9 consists of two components. The first component is the potential energy, which is equivalent to the Dirichlet energy in [1] because they both measure how much the signal varies across the graph. While [1] defines the Dirichlet energy by summing for the edges $(u,v)\in E$, the feature distance between nodes $u$ and $v$, in our case, it is defined by summing, for each node $v$, the feature difference between $v$ and neighboring nodes $u\in\mathcal{N}(v)$, therefore if and only if $(u,v)\in E$. In addition, our definition includes a second term, which represents the kinetic energy of the wave, capturing how node states change over time. This corresponds to the definition of energy for the propagation of waves in graphs [2]. From a physics perspective, kinetic energy reflects how fast a wave vibrates, while potential energy reflects the tension in the system. By keeping the energy in Eq. 9 constant, we ensure that node information is preserved during propagation. Thus, we can mitigate the problem of long-range propagation without focusing solely on oversmoothing phenomena. We would be happy to include this discussion in the revised paper if the reviewer agrees that it is important.
>
> [1] Di Giovanni, Francesco, et al. "Understanding convolution on graphs via energies." arXiv:2206.10991 (2022).
>
> [2] Joel Friedman and Jean-Pierre Tillich. Wave equations for graphs and the edge-based Laplacian. Pacific J. Math., 2004.
>
> 2) We thank the Reviewer for the question, though we are unsure whether we have interpreted it as intended. To this end, we would like to first elaborate on the nature of the tasks, in particular those in Section 4.1, which are intrinsically related to shortest path computation as they require computing the graph diameter, sssp, and node eccentricities. The goal of these kinds of tasks is to compute for each node $v$ in the graph its distance from the other nodes. This problem is fundamentally linked with information propagation, as node signals must travel from each node $v$ across the entire graph. Therefore, it requires nodes to iteratively exchange information with increasingly distant nodes. This process shares similarities with classical algorithms like Bellman-Ford or Dijkstra's algorithm. Indeed, Dijkstra's algorithm works by progressively expanding a frontier of visited nodes, where each node updates the shortest known distance to its neighbors.
> A standard GCN fails in solving these tasks since it performs a dissipative signal propagation as each node’s state is averaged with its neighborhood, causing rapid oversmoothing as depth increases. As a result, information about distant nodes becomes increasingly indistinguishable and ultimately lost. In contrast, our SONAR successfully solves these tasks because it performs a non-dissipative propagation. This is enabled by explicitly preserving the energy of the system (eq 9) in its design, with the result of preserving signals during propagation (as theoretically discussed in Section 3). This theoretical insight is empirically supported by the specifically designed long-range experiments in Sec 4.1 and 4.2, where our SONAR’s performance is orders of magnitude better than GCN, and by the sensitivity ablation study reported in our previous response. Indeed, such analysis shows how our SONAR can propagate information consistently even at long distances, while GCN falls short in the propagation.
> In the following, we also report an ablation (extracted from Table 8) that compares GCN (which can be considered as SONAR without energy preservation, adaptive resistance, and external forces) with our SONAR (with different combinations of its components).
>
> | Energy preservation | Adaptive Resistace | Dissipation | External Forcing | Line 50 | SSSP |
> |:---:|:---:|:---:|:---:|:---:|:---:|
> ||||| $\log_{10}$MSE $\downarrow$ | $\log_{10}$MSE $\downarrow$ |
> |  x  |  x  |  x  |  x  | **-6.4069 ±  0.4653** | -3.0213 ±  0.2431 |
> |  x  |  x  |  x  | $-$ | -5.7542 ±  0.6264   | -3.8182 ±  0.2239 |
> |  x  |  x  | $-$ |  x  | -6.2545 ±  0.3354   | -2.7857 ±  0.1731 |
> |  x  |  x  | $-$ | $-$ | -5.3827 ±  0.4078   | -2.8108 ±  0.2010 |
> |  x  | $-$ |  x  |  x  | -6.3598 ±  0.5403   | **-6.7515 ± 0.0589** |
> |  x  | $-$ |  x  | $-$ | -5.9058 ±  0.5576   | -6.5939 ±  0.2925 |
> |  x  | $-$ | $-$ |  x  | -4.9749 ±  0.2777   | -6.5241 ±  0.5738 |
> |  x  | $-$ | $-$ | $-$ | -5.4487 ±  0.7370 | -6.3226 ±  0.1126 |
> |   |   |   |   |  | |
> | $-$ | $-$ | $-$ | $-$ | -3.5032 ± 0.0001 | 0.9499 ± 0.0001 |
>
> We note that the last line of the table corresponds to GCN. As it is shown in the table, energy preservation is fundamental to achieving effective long-range propagation, hence strong performance in these tasks.
> We hope this response has clarified the Reviewer's question, and we would be happy to further engage in discussion if needed.

---

> ### Author Response · Authors · 2025-08-04
> **Response to further questions (2/2)**
>
> 3) We thank the reviewer for finding our sensitivity analysis useful, and we will include it in the revised version of the paper as suggested. In our previous answer, we reported the sensitivity for the best number of iterations (i.e., layers) of our method. For the sake of completeness, in the following, we report the full list of sensitivity norms with respect to the distance between the considered nodes and the number of iterations of the method.
>
>    | distance ↓ / num_iters → | 25 | 50 | 75 | 100 | distance ↓ / num_iters→ | 25 | 50 | 75 | 100 |
>    |---|:---:|:---:|:---:|:---:|---|:---:|:---:|:---:|:---:|
>    | **SONAR** | || | | **GCN** | $\times 10^{-5}$ | $\times 10^{-5}$ | $\times 10^{-5}$ | $\times 10^{-5}$ |
>    | 10 | 0.0115 | 0.0413 | 0.573 | 1.955 | 10 | 0.7078 | 0.0 | 1.1470 | 0.3562 |
>    | 20 | 0.00373 | 0.0334 | 0.703 | 5.084 | 20 | 0.0 | 0.0 | 1.0490 | 0.2980 |
>    | 30 | 0.0273 | 0.0616 | 1.269 | 4.341 | 30 | 0.3725 | 0.9779 | 0.5259 | 0.07450 |
>    | 40 | 0.0131 | 0.0253 | 0.584 | 1.061 | 40 | 0.0 | 0.0 | 0.0 | 0.0 |
>    | 50 | 0.0001862 | 0.0324 | 0.143 | 1.004 | 50 | 0.0 | 0.0 | 0.0 | 0.0 |
>
>    Even though we reported only the best configuration (as the sensitivity depends on multiple hyperparameters, e.g., number of iterations and step size), we note SONAR can propagate information over long distances regardless of the configuration, while GCN falls short in propagation. Indeed, our results indicate that information is successfully transferred between distant nodes for both low and high iteration values, with the best performance resulting from selecting the right number of iterations. We recall that the above sensitivities are measured on “line 50” in the graph transfer tasks. In this case, it is required to transfer information from node 0 to node 50. To perform such a task, node 0 needs to have a high influence on node 50, as they respectively represent the source and target for the propagation of information.

---

> > ### Comment · Reviewer_gMHH · 2025-08-04
> >
> > Thank you for the detailed comments. I am overall satisifed with the response and will recommend acceptance of this paper.

---

### Decision · Program_Chairs · 2025-09-17

**Decision:**

Accept (poster)

**Comment:**

SONAR is a novel GNN architecture that models information flow on graphs as oscillations governed by the wave equation, addressing long-range information propagation. It integrates adaptive edge resistances and state-dependent external forces, theoretically ensuring energy conservation and stable information propagation over long distances. Strengths include its clear wave-based motivation, comprehensive theoretical derivation, strong empirical results, and sub-quadratic complexity. Initial weaknesses involved a perceived lack of qualitative insight into wave-like dynamics, the use of some benchmarks that were questioned regarding their long-range nature, and the observation that certain components' effects appeared inconsistent across datasets.

The paper was accepted because of its principled and mathematically sound approach to a fundamental problem in graph representation learning, supported by robust theoretical guarantees and compelling experimental evidence.

During the rebuttal, reviewer gMHH’s concerns about qualitative insight and benchmarks were addressed with sensitivity analysis demonstrating SONAR's consistent information propagation and the inclusion of new experiments. Reviewer jhtz’s queries about component consistency and comparison with relevant prior work were met with explanations of task-dependent utility and discussions on model complexity. Reviewer byZL’s concerns regarding sensitivity to discretization schemes and step size were resolved by confirming theoretical robustness and demonstrating empirical stability. The authors' comprehensive responses and additional analyses significantly strengthened the paper, leading to a favorable assessment by the reviewers.